# In-situ spectroscopic observation of dynamic-coupling oxygen on atomically dispersed iridium electrocatalyst for acidic water oxidation

Hui Su[1,4], Wanlin Zhou[1,4], Wu Zhou[2], Yuanli Li[1], Lirong Zheng[3], Hui Zhang[1], Meihuan Liu[1], Xiuxiu Zhang[1], Xuan Sun[1], Yanzhi Xu[1], Fengchun Hu[1], Jing Zhang[3], Tiandou Hu[3], Qinghua Liu [1✉] & Shiqiang Wei [1✉]

Uncovering the dynamics of active sites in the working conditions is crucial to realizing increased activity, enhanced stability and reduced cost of oxygen evolution reaction (OER) electrocatalysts in proton exchange membrane electrolytes. Herein, we identify at the atomic level potential-driven dynamic-coupling oxygen on atomically dispersed hetero-nitrogen-configured Ir sites (AD-HN-Ir) in the OER working conditions to successfully provide the atomically dispersed Ir electrocatalyst with ultrahigh electrochemical acidic OER activity. Using in-situ synchrotron radiation infrared and X-ray absorption spectroscopies, we directly observe that one oxygen atom is formed at the Ir active site with an O-hetero-Ir-$N_4$ structure as a more electrophilic active centre in the experiment, which effectively promotes the generation of key *OOH intermediates under working potentials; this process is favourable for the dissociation of $H_2O$ over Ir active sites and resistance to over-oxidation and dissolution of the active sites. The optimal AD-HN-Ir electrocatalyst delivers a large mass activity of 2860 A $g_{metal}^{-1}$ and a large turnover frequency of 5110 $h^{-1}$ at a low overpotential of 216 mV (10 mA cm$^{-2}$), 480–510 times larger than those of the commercial $IrO_2$. More importantly, the AD-HN-Ir electrocatalyst shows no evident deactivation after continuous 100 h OER operation in an acidic medium.

[1] National Synchrotron Radiation Laboratory, University of Science and Technology of China, Hefei 230029 Anhui, P. R. China. [2] School of Chemistry and Chemical Engineering, Key Laboratory for Green Processing of Chemical Engineering of Xinjiang Bingtuan, Shihezi University, Shihezi 832003, China. [3] Beijing Synchrotron Radiation Facility, Institute of High Energy Physics, Chinese Academy of Sciences, Beijing 100049, China. [4]These authors contributed equally: Hui Su, Wanlin Zhou. ✉email: qhliu@ustc.edu.cn; sqwei@ustc.edu.cn

Polymer electrolyte membrane water electrolysis (PEMEC) is viewed as a clean and promising way to convert electrical energy to hydrogen, playing an important role in modern sustainable energy conversion and storage devices[1–3]. Due to the sluggish oxygen evolution reaction (OER) kinetics and harsh acidic environment, the PEMEC electrode should be catalytically active and highly stable under operating conditions[3–5]. Currently, iridium-based metal oxides, such as $IrO_2$, have been widely used in water electrolysis[6,7]. Unfortunately, $IrO_2$ still tends to decompose in strongly acidic media ascribed to the loss of active surface area and the changes in iridium species oxidation state under prolonged operating conditions[8,9]. Although significant efforts have been devoted to developing various acidic electrocatalysts, the corresponding catalytic mechanisms are still unclear due to the complex reaction system[10,11]. Furthermore, most electrodes employed in the in situ experiments are fabricated by simply casting electrocatalyst powder onto substrates with only physical contact, and the detached electrode sample affects the signal of the in situ experiment, leading to inauthentic evolution under OER conditions. Therefore, the development of highly acid-resistant OER electrocatalysts with excellent catalytic activity and a stable electrode configuration from the aspect of mechanistic understanding is anticipated for practical PEMEC applications but remains a considerable challenge. To this end, atomic-level identification of the nature of the active site under working conditions is imperative to design efficient acid-resistant OER electrocatalysts.

To date, enormous efforts have been made to improve acidic OER electrocatalysts, such as transition metal substitution, structural engineering, and element doping (Co-RuIr, $Ir_xCu$, $IrO_x/SrIrO_3$, and Ir-STO)[12–15]. However, the changes in oxidation state of the Ir-based samples would cause continuous dissolution of the active metal attributed to the presence of manifold Ir–O coordination[9]. Recently, downsizing expensive $IrO_x$ to atomically dispersed iridium has provided an efficient pathway to create cost-efficient electrocatalysts and improve their activity and stability because of the strong metal–support interaction[16]. The electrocatalytic activity and stability of the atomically dispersed electrocatalysts are highly dependent on the local coordination structure of metal sites, which is directly related to the interaction between electrons and the geometric structure. Strong metal–support interactions can facilitate large electron transfer from the metal to neighboring atoms towards high acidic OER activity. Although various $M–N_x$ moieties have been achieved in the atomically dispersed electrocatalysts for efficient electrocatalytic reactions[17–19], inadequate electronic structure activation and unreliable atomic configurations limit the potential of M–N–C electrocatalysts for high efficiency and stable acidic water oxidation[20,21]. Hence, it is urgently necessary to find an effective way to activate the acidic OER activity and improve the dissolution resistance of ultralow iridium electrocatalysts via strong electron coupling between the atomically dispersed active sites and their metal anchoring substrate[22]. Moreover, it has been reported that the structure of atomically dispersed electrocatalysts is easily reconstructed to a higher valence state for faster adsorption of reactants to achieve an efficient activity in electrocatalytic reactions. This means that the electronic structure of the active center can be manipulated at low potentials to significantly improve the electrocatalytic activity. Therefore, for the atomically dispersed metal centers embedded into 3D conductive substrates with stable active site configurations, an in-depth understanding of the reaction dynamics of active electrocatalytic centers at the atomic scale can provide deep insights into the design of acidic OER electrocatalysts.

Here, to realize high electrocatalytic activity and long-term durability in acidic media, we designed a new type of atomically dispersed Ir active sites coupled to a hetero-nitrogen-configured 3D carbon substrate as ultralow-iridium electrocatalysts via a controllable "electric-driven amino-induced" strategy. At the atomic level, the active hetero–Ir–$N_4$ moieties were spatially confined in the hetero-nitrogen-configured 3D carbon substrate with strong interfacial chemical coupling owing to the selective ion-bonding effect in the weak redox units of PAni and electric-driven $NH_2$ (Fig. 1a). Interestingly, in situ X-ray absorption fine structure (XAFS) spectroscopy revealed that one oxygen atom was formed on the Ir active site in the form of an O–hetero–Ir–$N_4$ moiety under a low driving potential, which accelerated the transfer of electrons from the metal sites to neighboring atoms toward faster reaction kinetics. More importantly, in situ synchrotron radiation infrared (SRIR) and electrochemical impedance spectroscopy (EIS) directly observed $H_2O$ adsorption under low driving voltage and crucial *OOH intermediate production on the O–hetero–Ir–$N_4$ active sites because of the electrophilic effect of coupled oxygen during the acidic OER process, quickly overcoming the rate-determining step of water oxidation and then boosting efficient acidic OER activity and stability. As a result, the developed AD–HN–Ir electrocatalyst delivers a low overpotential of 216 mV to achieve a current density of 10 mA cm$^{-2}$ for acidic OER with an ultra-high mass activity of 2860 A g$_{metal}$$^{-1}$ and a high turnover frequency (TOF) of 5110 h$^{-1}$. Furthermore, the AD–HN–Ir electrocatalyst showed ~97% Faradaic efficiency for the OER and had no obvious attenuation of acidic OER activity in continuous 100 h operation in an acidic medium.

## Results and discussion

**Structure characterization of AD–HN–Ir electrocatalyst.** An ultralow-iridium electrocatalyst with atomically dispersed Ir active sites coupled to a hetero-nitrogen-configured 3D carbon substrate was synthesized via a controllable "electric-driven amino-induced" strategy. The polyaniline layer with plenty of surface reductive benzenoid-amine groups was first electrodeposited onto a 3D carbon substrate and then functionalized with amino groups by mild heat treatment in ammonia solution. It is noted that the surface reductive benzenoid-amine groups and -$NH_2$-derived uncoordinated N sites act as anchoring sites for Ir atoms in the $H_2IrCl_4$ solution via a typical ion exchange strategy. Notably, the ultralow iridium atoms were chemically coupled with N sites on the hetero-nitrogen-configured 3D carbon substrate to form the active and stable AD–HN–Ir electrocatalyst. The obtained AD–HN–Ir electrocatalyst can be verified by scanning electron microscopy (SEM, Supplementary Figs. 1–3) and transmission electron microscopy (TEM, Supplementary Fig. 4). The SEM and TEM images (Fig. 1b, c) clearly show that the atomically dispersed Ir atoms chemically coupled onto the hetero-nitrogen-configured 3D carbon substrate, with no noticeable metal particles observed. In particular, high-angle annular dark-field scanning TEM (HAADF-STEM) further confirmed that atomically dispersed Ir atoms were riveted on the hetero-nitrogen-configured 3D carbon substrate surface with a particle size of ~0.21–0.23 nm (Supplementary Fig. 5). The elemental mappings (Fig. 1d) indicate the uniform distribution of C, N, and Ir in the selected carbon fiber area. Similarly, coupled plasma optical emission spectrometry (ICP-OES) characterizations reveal the existence of Ir in the obtained AD–HN–Ir electrocatalyst with an ultralow content of approximately 3.5 μg cm$^{-2}$. X-ray diffraction results (Supplementary Fig. 6) further showed that no Ir nanoparticles were observed for AD–HN–Ir electrocatalyst, indicating a uniform dispersion of Ir active sites throughout the hetero-nitrogen-configured 3D carbon substrate.

To determine the local structure of Ir atoms at the atomic scale, X-ray absorption near-edge spectroscopy (XANES) of the C and N K-edge was performed. The C K-edge spectrum of NC (PAni

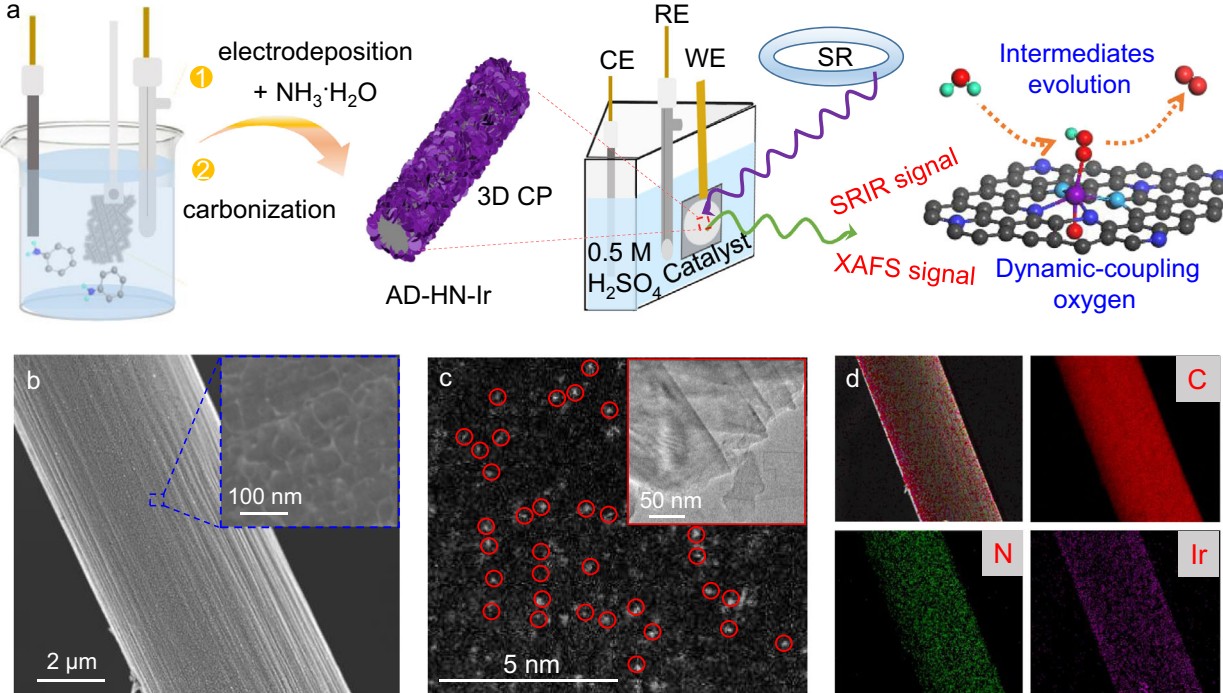

**Fig. 1 In situ electrochemical measurement and structural characterizations of AD–HN–Ir electrocatalyst. a** Scheme of the synthetic process. CE counter electrode, WE, working electrode, RE reference electrode, CP carbon paper, SR synchrotron radiation, C black, $N_1$ royal blue, $N_2$ indigo, O red, H green, Ir purple. **b** SEM image of AD–HN–Ir electrocatalyst, and the inset is a partially enlarged image. **c** HAADF-TEM image and the inset is a TEM image. **d** STEM-EDS mapping images for AD–HN–Ir electrocatalyst.

annealed at 800 °C) in Fig. 2a shows two obvious peaks located at 285.4 and 288.7 eV that can be assigned to the $1s \rightarrow 2p$ excitation of C–C $\pi^*$ and C–$N_2$, respectively[23]. After Ir active sites coupled onto the NC substrate, a new peak was observed at 287.8 eV, which can be attributed to the excitation of C–$N_1$, suggesting the presence of hetero-nitrogen atoms after ammonia water treatment. The N $K$-edge spectrum in Fig. 2b shows the excitation peaks of C–N and hetero-Ir–$N_x$. The high-energy shift of C–N reveals that the introduction of amino-derived N results in electron transfer between C and N. The above results clearly reveal the coordination of atomically dispersed Ir active sites with heteropyridinic- and amino-derived nitrogen atoms for AD–HN–Ir electrocatalyst. Furthermore, the five fitting peaks of 406.3, 401.2, 400.1, 399.7, 398.9, and 398.4 eV in the N $1s$ XPS spectra (Fig. 2c) can be assigned to N–O, graphitic N, pyrrolic N, Ir–N, Ir–amino–N and pyridinic N, respectively[24,25]. This further demonstrates the formation of hetero-nitrogen-configured Ir active sites for AD–HN–Ir electrocatalyst. Moreover, the Ir $4f$ XPS spectra clearly reveal that a high-valence Ir atom is coupled onto the carbon substrate in the form of $Ir^{3+}$ with a hetero-N configuration (Fig. 2d)[26]. The atomic structure configuration of the Ir site was obtained by measurements of Ir $L_3$-edge XAFS spectra. As seen from the XANES of the Ir $L_3$-edge in Fig. 2e, it can be inferred that the high-valence Ir atom forms the coordination structure of Ir–N based on the position of the absorption edge and the intensity and width of the white line peak. Figure 2f shows one main peak located at ~1.6 Å that can be assigned to the first shell of Ir–N coordination according to the N $1s$ XPS result as well as the electrophilic center of N[27]. The $L_3$-edge EXAFS $k^2\chi(k)$ of the AD–HN–Ir electrocatalyst (Supplementary Fig. 8) shows that the atomically dispersed Ir active sites with Ir–N coordination configuration in the first shell was anchored to a hetero-nitrogen-configured 3D carbon substrate. The peaks at higher R can be attributed to higher frequency noise

and a small amount (10%) of Ir–Ir bonds because of the low Ir content for fluorescence XAFS measurement. To further quantify the local coordination structure of Ir, the EXAFS curves of the Ir atom were fitted for AD–HN–Ir electrocatalyst. The fitting curve of the $k^2$-weighted EXAFS spectrum (Fig. 2f) and corresponding structure parameters in Supplementary Table 1 show that the first-shell coordination number is close to 4 with two short and two long bonds, consisting of first-shell of Ir–$N_1$ (1.92 Å) and Ir–$N_2$ (1.95 Å) coordination, suggesting the hetero-N configuration of atomically dispersed Ir active sites. Therefore, the above results clearly verify the formation of hetero–Ir–$N_4$ moieties coupled onto the hetero-nitrogen-configured 3D carbon substrate for the AD–HN–Ir electrocatalyst.

**OER performance of AD–HN–Ir electrocatalyst.** The electrocatalytic OER activities of AD–HN–Ir electrocatalyst were determined in 0.5 M $H_2SO_4$ with a three-electrode system, along with Ir–NC, Ir nanoparticle loading NC electrocatalyst (Ir–NP/NC), and commercial $IrO_2$ for comparison. Figure 3a shows the linear sweep voltammetry curves for the AD–HN–Ir electrocatalyst and reference samples. Dramatically, AD–HN–Ir electrocatalyst requires ultralow overpotentials of only 216 and 292 mV to achieve water oxidation current densities of 10 and 100 mA cm$^{-2}$, respectively. In contrast, as shown in Fig. 3b, Ir–NC electrocatalyst and commercial $IrO_2$ display insufficient acidic OER activity (320 and 412 mV and 300 and 385 mV, respectively, Supplementary Fig. 9), suggesting that hetero-nitrogen-configured Ir active sites achieve efficient water oxidation activity in acidic media by functionalization of polyaniline and amino groups. The morphology structure and OER performance of Ir electrocatalysts at different pyrolysis temperatures were shown in Supplementary Fig. 10, revealing the AD–HN–Ir (800 °C) electrocatalyst with better carbonization degree and

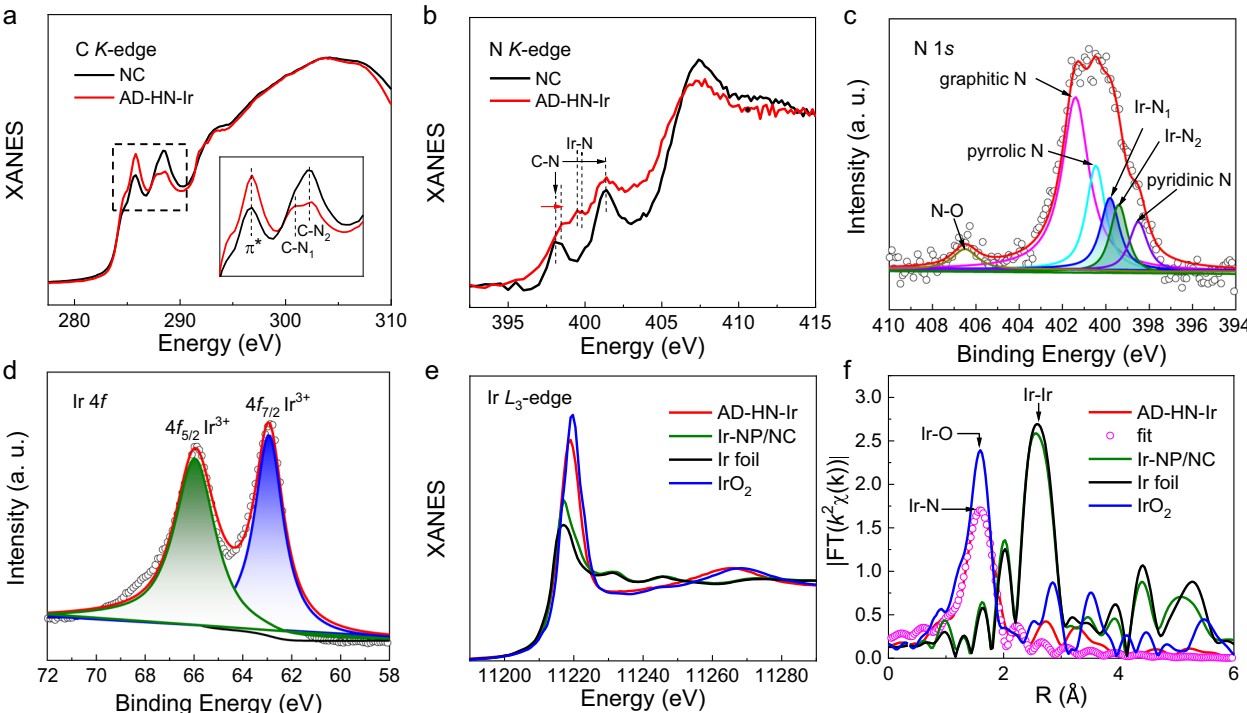

**Fig. 2 Electronic structural characterizations of AD–HN–Ir electrocatalyst. a** C and **b** N K-edge XANES spectra of NC and AD–HN–Ir electrocatalysts. **c** N 1s and **d** Ir 4f XPS spectra of AD–HN–Ir electrocatalyst. **e** Ir $L_3$-edge XANES spectra and **f** Fourier transforms (FTs) of the Ir $L_3$-edge EXAFS oscillations of AD–HN–Ir, Ir-NP/NC, Ir foil, and $IrO_2$ and the fitting curve of $k^2$-weighted EXAFS spectrum of AD–HN–Ir electrocatalyst.

atomically dispersed Ir active sites. Considering the effect of temperature on acidic OER performance, we conducted a water-splitting test under different operating temperatures. The OER activity is quite temperature-dependent, and the overpotentials of AD–HN–Ir electrocatalyst decrease from 216 to 204 mV at a current density of 10 mA cm$^{-2}$ with the increase of temperature to 80 °C, which is consistent with what has been observed that a higher temperature provides faster OER kinetics and better OER activity (Supplementary Fig. 11). Importantly, the electrochemically active surface area (ECSA) of AD–HN–Ir electrocatalyst was obtained by the roughness factor (Supplementary Figs. 12 and 13), and the specific activity of AD–HN–Ir electrocatalyst still surpassed those of Ir–NC electrocatalyst and $IrO_2$ when normalizing the current density to per ECSA (Supplementary Fig. 14). Moreover, Fig. 3c displays a smaller Tafel slope of 39 mV dec$^{-1}$, suggesting a faster OER kinetics and electron transfer over the Ir active sites. Based on the Arrhenius plot results of Fig. 3d and Supplementary Figs. 15 and 16, the activation energy of AD–HN–Ir electrocatalyst is downshifted to 26.56 kJ mol$^{-1}$ relative to that of commercial $IrO_2$ (44.27 kJ mol$^{-1}$) under an overpotential of 216 mV, which further proves the faster OER kinetics after integrating high-valence atomically dispersed Ir active sites onto the hetero-nitrogen-configured 3D carbon substrate.

To further obtain the intrinsic activity of AD–HN–Ir electrocatalyst, the mass activity, and TOF were calculated according to the Ir active sites (Fig. 3e). Notably, the mass activity and TOF of the AD–HN–Ir electrocatalyst are approximately 2860 A g$_{metal}^{-1}$ and 5110 h$^{-1}$ at a small overpotential of 216 mV, ~480 and 510 times larger than those of the commercial $IrO_2$ (6 A g$_{metal}^{-1}$, 10 h$^{-1}$) by performance test statistics. Notably, the AD–HN–Ir electrocatalyst displays an ultrahigh mass activity of 10,900 A g$_{metal}^{-1}$ and TOF of 18,900 h$^{-1}$ at a representative overpotential of 260 mV. To the best of our knowledge, the mass activity of the AD–HN–Ir electrocatalyst is outstanding among the reported acidic OER electrocatalysts

(Supplementary Table 2). With increasing overpotentials, the mass activity and TOF increase significantly, indicating faster OER catalytic kinetics. The details of the oxygen product quantification by gas chromatography are presented in Supplementary Fig. 17. The AD–HN–Ir electrocatalyst offers an excellent four-electron (4e$^{-}$) OER with Faradaic efficiencies of 95–98% under various potentials in an acidic medium. Due to the excellent acidic OER performance of AD–HN–Ir electrocatalyst, we conducted an overall water splitting test in a two-electrode configuration. The AD–HN–Ir electrocatalyst and commercial Pt/C were used as the anode and cathode, respectively. Supplementary Fig. 18 shows that the input potential of water splitting in practical electrolyzers is approximately 1.49 V for achieving a current density of 10 mA cm$^{-2}$, which is significantly superior to that of standard electrodes (Pt/C vs. $IrO_2$, 1.58 V). The stability of electrocatalyst is an important performance index for evaluating the real application potential, especially in harsh operating conditions. To assess the durability of AD–HN–Ir electrocatalyst and the reference sample, a chronopotentiometry test at a current density of 10 mA cm$^{-2}$ was performed. As shown in Fig. 3f, the acidic OER activity has no apparent attenuation in the continuous operation for 100 h for the AD–HN–Ir electrocatalyst, obviously outperforming commercial $IrO_2$, which is attributed to the formation of hetero–Ir–N$_4$ moieties strongly coupled onto the hetero-nitrogen-configured 3D carbon substrate (Supplementary Figs. 19 and 20). Interestingly, the AD–HN–Ir electrocatalyst delivers good activity and stability at a high current density of 100 mA cm$^{-2}$ (Supplementary Fig. 21a, b). This electrocatalyst can efficiently avoid repeated changes of the Ir oxidation state for faster water oxidation kinetics and strong acid resistance. In summary, the high durability of AD–HN–Ir electrocatalyst under continuous operating conditions is attributed to the stable hetero–Ir–N$_4$ moieties, which can inhibit the peroxidation and dissolution of Ir active sites. To further clarify the activity and stability at higher current densities, a higher current density of 500 mA cm$^{-2}$ was tested for the AD–HN–Ir electrocatalyst. The AD–HN–Ir electrocatalyst exhibits poor stability with

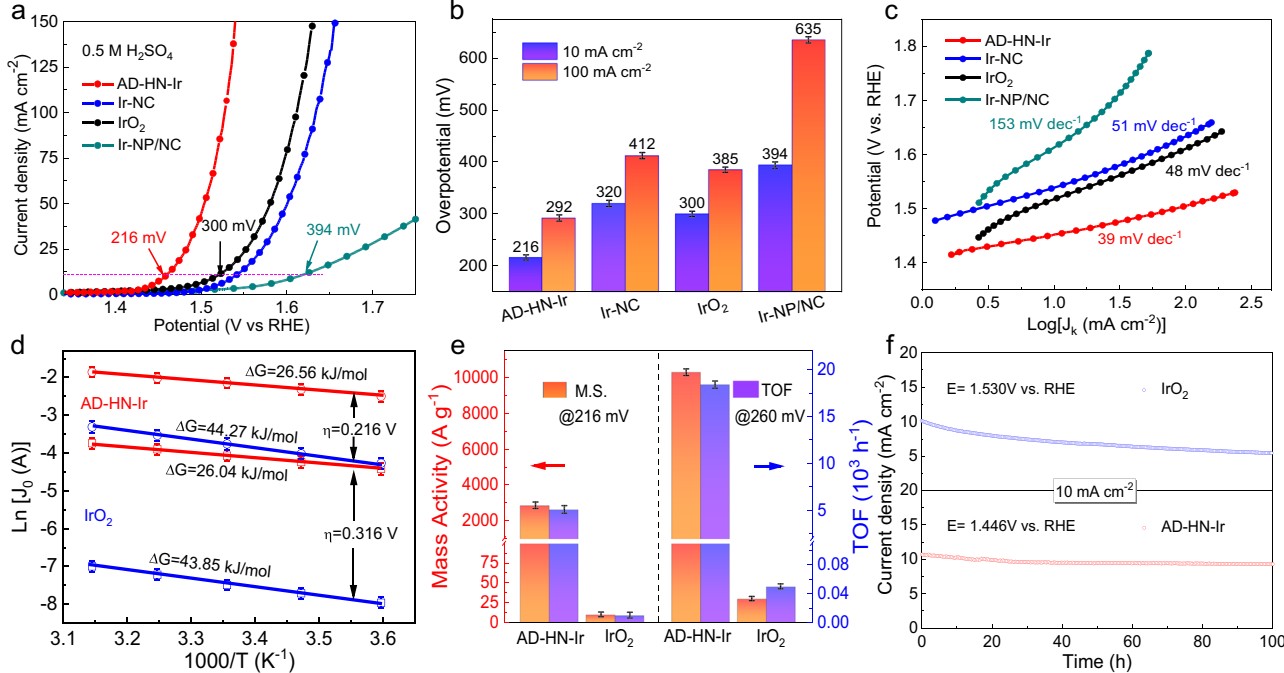

**Fig. 3 Electrochemical OER properties. a** Linear sweep voltammetry (LSV) curves. **b** Overpotentials at 10 and 100 mA cm$^{-2}$ and **c** Tafel slopes for AD–HN–Ir, Ir–NC, IrO$_2$, and Ir–NP/NC. **d** Arrhenius plots of AD–HN–Ir electrocatalyst. **e** Mass activity (M.A.) and turnover frequency (TOF) at overpotentials of 216 and 260 mV for AD–HN–Ir electrocatalyst and IrO$_2$. **f** OER stability for AD–HN–Ir electrocatalyst and IrO$_2$ at applied potentials of 1.446 and 1.530 V, respectively.

retention of ~50% of the initial current density (Supplementary Fig. 21c and 21d). Significantly higher S-number and longer lifetime of AD–HN–Ir electrocatalyst were observed under a lower potential of 1.53 V (reach to 100 mA cm$^{-2}$) (the subsequent potentials are all relative to a reversible hydrogen electrode), suggesting that the decay of the stability was attributed to the corrosion of the carbon substrate at a high potential (Supplementary Fig. 22)[28]. It is difficult to achieve a high activity and stability at higher current densities of 1000 and 2000 mA cm$^{-2}$ because of oxidation corrosion of the carbon substrate under high oxidation potentials. To capture the industrial potential of the AD–HN–Ir electrocatalyst, we also carried out the stability of the AD–HN–Ir electrocatalyst at high current density by a PEM electrolyzer system under simulated industrial conditions (80 °C, Supplementary Fig. 23). The resultant electrolyzer delivered 500 mA cm$^{-2}$ at ~1.75–1.85 V over 10 h operation, indicating a degree of stability of the AD–HN–Ir electrocatalyst. However, the stability of the electrocatalyst decreased significantly at a high current of 1000 mA cm$^{-2}$, which is attributed to the significant oxidation of the carbon substrate.

**In situ EIS and SRIR analysis.** EIS is an effective electrochemical measurement method to obtain the adsorption and desorption kinetics information of the reactants on the electrode surface. To intuitively obtain the evolution of the oxygen-containing reactive species over Ir active sites, in situ EIS tests were performed at different applied potentials to further understand the electrochemical reaction kinetics process[29]. As seen from the Nyquist plots in Fig. 4a, b, the measured impedance of the acidic OER process on AD–HN–Ir and Ir–NC electrocatalysts was measured from the open circuit potential (OCP, ~0.95 V) to 1.45 V. Through the direct changing trend and size of the semicircle in the Nyquist plots, faster kinetics and adsorption of reactants with increasing potentials can be revealed. At the same time, AD–HN–Ir electrocatalyst shows faster reaction kinetics, which is attributed to active hetero–Ir–N$_4$ coordination and the strong

coupling substrate effect. Importantly, these Nyquist plots should first need to be fitted by equivalent circuit diagrams to assess the key fitting parameters $R_{ct}$ and $C_{dl}$ (Supplementary Fig. 24). $R_{ct}$ and $C_{dl}$ can be used to represent the oxygen-containing reactive species ion adsorption resistance and pseudocapacitance on the catalyst surface. It can be seen from the changes of fitting results for $R_{ct}$ in Fig. 4c that the $R_{ct}$ of AD–HN–Ir electrocatalyst is lower than that of Ir–NC, suggesting faster adsorption kinetics of key intermediates oxygen-containing reactive species during the acidic OER process (Supplementary Tables 3 and 4). Notably, $R_{ct}$ decreases faster than that of Ir–NC within 1.1 V, which further reveals that oxygen species are rapidly adsorbed at low driving potentials to produce key intermediates oxygen-containing reactive species accumulated over the active hetero–Ir–N$_4$ moieties.

To effectively identify the reaction intermediates and access the catalytic reaction mechanism, in situ SRIR measurements, which are highly sensitive to oxygen-containing intermediates, were performed with a homemade electrochemical SRIR cell[30,31]. The SRIR results of AD–HN–Ir electrocatalyst were obtained at different applied potentials, as shown in Fig. 4d. A new absorption band at 784 cm$^{-1}$ was observed for AD–HN–Ir electrocatalyst at a potential of 1.25 V, which can be assigned to the emergence of Ir–O during the OER process[32]. This indicates structural self-optimization of the Ir site under a low driving potential to form the active structure O–hetero–Ir–N$_4$ by oxygen coupling. The intensity of vibration absorption shows a positive correlation with a further increase in applied potentials and remains unchanged when the potential exceeds 1.35 V. A new absorption band at a vibration frequency of 1055 cm$^{-1}$ was observed for AD–HN–Ir electrocatalyst when the potential >1.35 V, which could be attributed to the emergence of crucial intermediate *O–OH during the acidic OER process[31]. In comparison, in situ SRIR measurements were performed for Ir–NC at typical potentials of 1.30 V (representing initial potential) and 1.45 V (representing reaction potential). Figure 4e shows that no obvious absorption bands were observed at

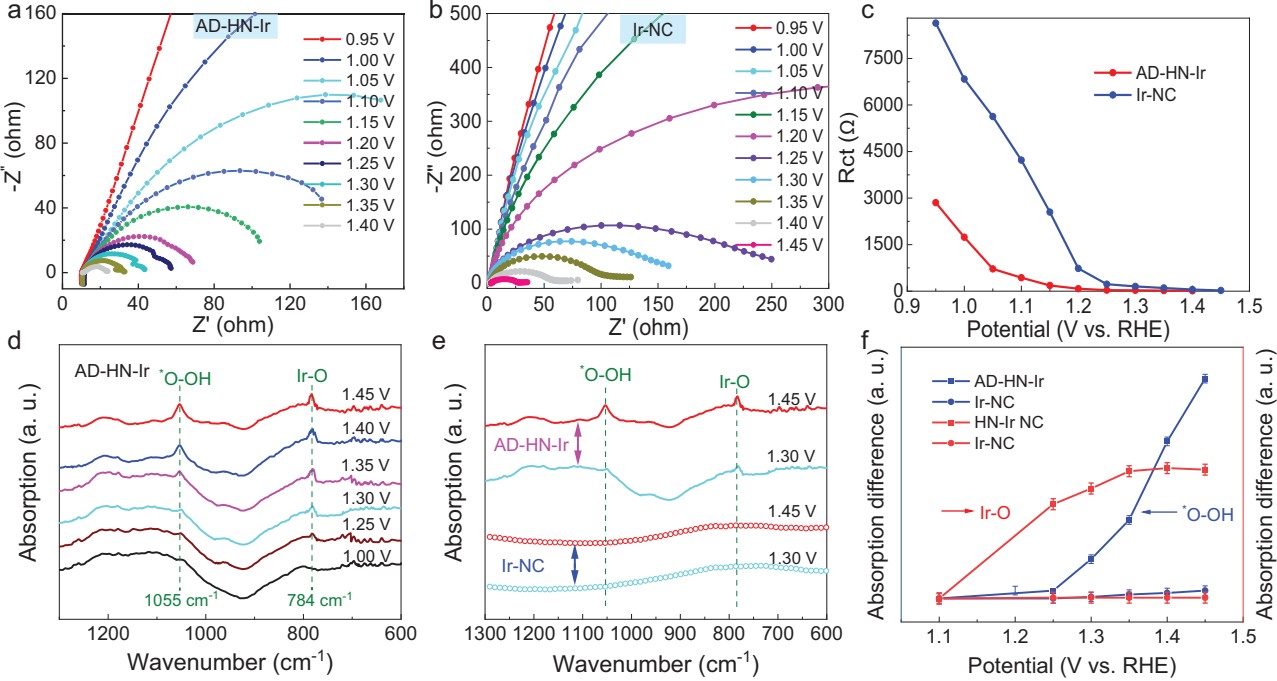

**Fig. 4 In situ EIS and SRIR measurements.** Nyquist plots for **a** AD–HN-Ir and **b** Ir-NC electrocatalysts at different applied potentials in 0.5 M $H_2SO_4$. **c** Response of the ion adsorption resistance ($R_{ct}$) at different potentials for AD–HN-Ir and Ir-NC electrocatalysts. **d** In situ SRIR measurements in the range of 1300–600 cm$^{-1}$ under various potentials for AD–HN-Ir electrocatalyst during the OER process. **e** In situ SRIR results in the range of 1300–600 cm$^{-1}$ at typical potentials of 1.30 and 1.45 V for AD–HN-Ir and Ir-NC electrocatalysts. **f** Intensity difference of the infrared signals at 1055 and 784 cm$^{-1}$ versus potentials for AD–HN-Ir and Ir-NC electrocatalysts during the OER process.

vibration frequencies of 1300–600 cm$^{-1}$ for Ir–NC under a potential of 1.45 V (Supplementary Fig. 26). This phenomenon indicates that high-valence Ir active sites coupled onto the hetero-nitrogen-configured 3D carbon substrate by a strong coupling substrate effect can preadsorbed oxygen to form a stable and highly active O–hetero–Ir–N$_4$ structure, obviously accelerating the four-electron kinetics process and promoting the formation of key OER reaction intermediates. To clarify the dynamic relationships between Ir–O and intermediates *OOH, the peak intensity enhancements of the absorption bands at 784 and 1055 cm$^{-1}$ were plotted as a function of applied potential in Fig. 4f. Notably, along with the increased applied potentials, the vibration band intensity of the key *OOH intermediate for the AD–HN–Ir electrocatalyst increases obviously more than that of Ir–NC. Most importantly, the coupling of an oxygen atom occurs at 1.25 V, clearly before the appearance of a new vibration band of the key *OOH intermediate for AD–HN–Ir electrocatalyst under OER working conditions. The dynamically coupled O disappears after reaction by the SRIR results (Supplementary Fig. 27), suggesting that the coupling of one oxygen atom on the Ir active site is a dynamic process with the change of potentials. The above results clearly reveal that the potential-driven coupling of oxygen over the Ir active sites was observed, which would in turn evidently promote the generation of key *OOH intermediates over the O–hetero–Ir–N$_4$ moieties towards an efficient and fast 4e$^-$ OER process in the acidic medium.

**In situ XAFS analysis.** To further assess the evolutionary nature of the atomically dispersed Ir active sites under working conditions, in situ XAFS measurements (Supplementary Fig. 28) were carried out in a homemade cell by using a three-electrode standard electrochemical workstation[33]. The Ir $L_3$-edge XANES spectra under different potentials in Fig. 5a show that the intensity of the white line of

AD–HN–Ir electrocatalyst increases gradually with increasing potentials, accompanied by a slight positive shift. This indicates that more electron vacant states of Ir active sites were observed under potential-driven conditions, where more electrons moved from Ir to nearby atoms and adsorbed oxygen species to active electronic surface states of Ir centers. This result suggests that the dynamically coupled oxygen on the Ir active site under potential-driven conditions obviously optimized the electronic structure of hetero-Ir–N$_4$ moieties for faster reaction kinetics. To further clarify the evolution of the local coordination structure of Ir active sites under the applied potential, FT of the Ir-$L_3$ edge EXAFS is shown in Fig. 5b, where a dominant peak of ~1.6 Å can be attributed to the first-shell of Ir–N/O/C coordination. Interestingly, the peak intensity increased significantly at a potential of 1.25 V and further enhanced with increased potential applied onto the electrode, suggesting local structural self-optimization and adsorption of reactive species under OER working conditions. The difference of $L$-edge EXAFS $k^2\chi(k)$ functions for the AD–HN–Ir electrocatalyst under different applied potentials further proves the dynamic evolution of Ir active sites during the OER process. Quantitatively, the Ir $L_3$-edge EXAFS fitting results (Fig. 5c, Supplementary Fig. 29 and Tables 5 and 6) clearly show an additional first-shell of Ir-O coordination with a bond length of 2.06 Å at an applied potential of 1.25 V. This result further proves that oxygen was coupled on the Ir active site in the form of an O–hetero–Ir–N$_4$ moiety with a coordination number of $N = 5$ (two short Ir–N bonds, two long Ir–N bonds and one Ir–O bond). With the applied potential increasing to 1.35 V, a new first-shell of Ir–O coordination was observed over the active O–hetero–Ir–N$_4$ moieties with a coordination number of two for the first-shell of Ir–O coordination. This result indicates that oxygen coupled onto Ir active sites can promote the adsorption of reactive oxygen species under the significant electrophilic effect of dynamic-coupling oxygen. With a further increase in the bias potential to 1.45 V, the FT peak intensity shows a slight increase accompanied by contraction of the first-shell

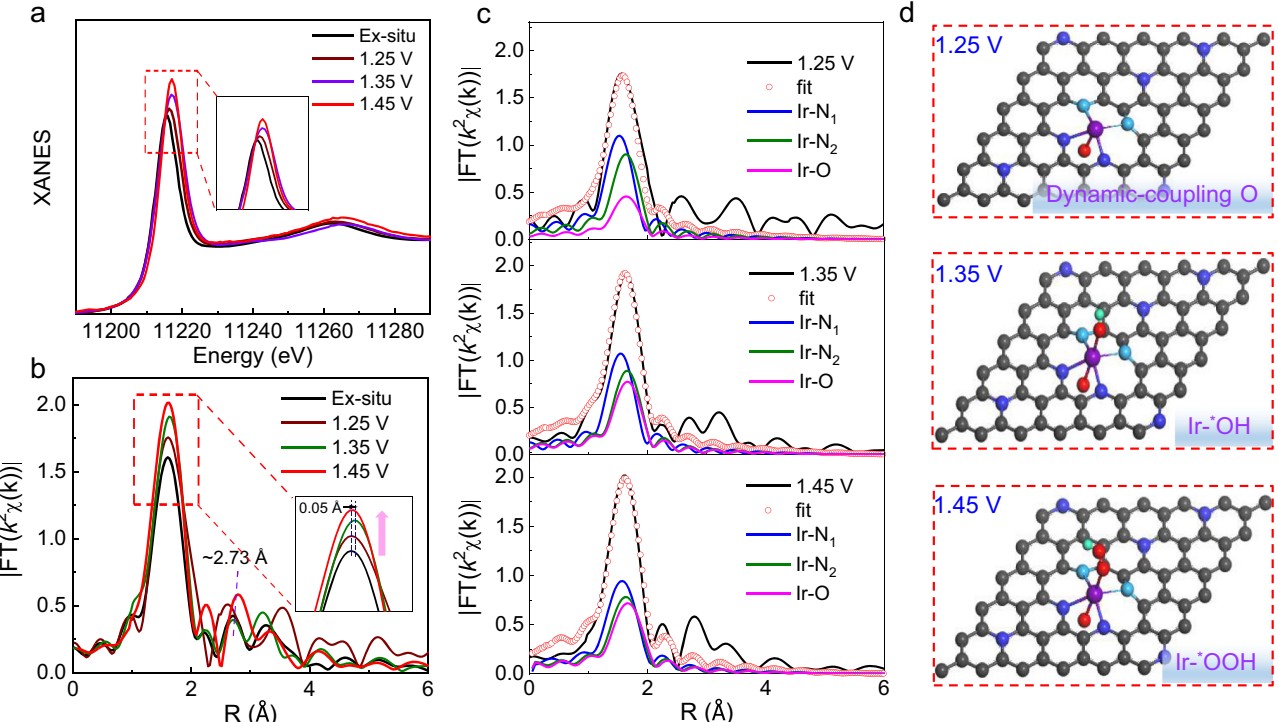

**Fig. 5 In situ XAFS measurement. a** In situ XANES spectra was recorded at the Ir $L_3$-edge of AD–HN–Ir electrocatalyst at different applied potentials from 1.25 to 1.45 V during the OER process. **b** Corresponding $k^2$-weighted Fourier transform (FT) spectra. **c** The fitting curves recorded at the Ir $L_3$-edge and **d** OER mechanism diagrams of AD–HN–Ir electrocatalyst at 1.25, 1.35, and 1.45 V. C, black; $N_1$, royal blue; $N_2$, indigo; O, red; H, green; Ir, purple.

of Ir–O coordination, indicating the deprotonation of adsorbed oxygen-species into the key *OOH intermediates on the activated O–hetero–Ir–$N_4$ moieties by electrophilic effects, which is consistent with the in situ SRIR results. Above all, the in situ SRIR and XAFS results jointly reveal that Ir active sites coupled onto the hetero-nitrogen-configured 3D carbon substrate by a strong coupling substrate effect can dynamically couple one oxygen atom to form a stable and highly active O–hetero–Ir–$N_4$ structure during the reaction process, obviously accelerating the four-electron kinetics process and promoting the formation of key OER reaction intermediates *OOH for efficient acidic-OER activity (Fig. 5d).

In summary, a highly efficient and durable AD–HN–Ir electro-catalyst with strong coupling hetero–Ir–$N_4$ moieties was developed by a controllable "electric-driven amino-induced" strategy. Dynamic mechanism studies using advanced in situ XAFS and SRIR techniques indicated that one oxygen atom is coupled onto the Ir active site and that a key *OOH intermediate is produced by the electrophilic effect of coupled oxygen over O–hetero–Ir–$N_4$ moieties under working potentials, which greatly accelerates the four-electron reaction kinetics towards efficient acidic OER activity and stability. The as-obtained AD–HN–Ir electrocatalyst delivers an ultralow overpotential of 216 mV at 10 mA cm$^{-2}$, reaching an ultrahigh mass activity of 2860 A g$^{-1}$ and a high TOF of 5110 h$^{-1}$.

## Methods

**Synthesis of AD–HN–Ir electrocatalyst.** The AD–HN–Ir electrocatalyst was fabricated in three typical well-designed steps, consisting of surface functionali-zation of 3D carbon paper (CP), surface amino treatment and ion exchange, and high-temperature pyrolysis. First, the 3D CP was activated and cleaned by calcining it at 500 °C under air for 2 h and then washed with concentrated nitric acid and deionized water. Next, polyaniline was electrodeposited on the 3D substrate in a three-electrode system. Specifically, pretreated CP, carbon rod, and saturated calomel electrode (SCE) were employed as working, counter, and reference elec-trodes, respectively, and a mixed solution of aniline (10 ml), concentrated nitric acid (12 ml), and deionized water (78 ml) was used as the electrolyte. Considering the avoidance of impurity doping of anions, we chose HNO$_3$ as an electrolyte in the electrodeposition process. The working electrode was first subjected to anodic

treatment at an oxidative constant voltage of 0.7 V (SCE) for 400 s to allow full polymerization of aniline molecules on the electrode surface, which was then switched to a reductive constant voltage of −0.3 V (SCE) to tune the ratio of quinonoid- and benzenoid-amine groups in the surface polyaniline layer to com-pete for the surface adsorption of NO$_3^-$. Subsequently, the obtained substrate was intruded into concentrated ammonia water for hydrothermal treatment for 5 h at 90 °C. Subsequently, the functionalized carbon substrate was immersed in a 60 ml of an aqueous solution containing H$_2$IrCl$_4$ (0.06 mmol/L) at 70 °C under con-tinuous stirring for 2 h, where anion exchange between [IrCl4]$^{2-}$ and preadsorbed NO$_3^-$ as well as [IrCl4]$^{2-}$ adsorption over the amino group was realized on the surface polyaniline layer of the carbon substrate. The surface benzenoid-amine and amino groups in polyaniline can easily couple with Ir complex ions, firmly anchoring Ir ions with heterogeneous nitrogen on the surface of the CP. Finally, the obtained 3D CP with rich Ir atoms was transferred to a tubular furnace for high-temperature annealing in an argon atmosphere, where the furnace was maintained at 800 °C for 3 h to obtain the AD–HN–Ir electrocatalyst. The ICP–OES char-acterizations reveal the existence of Ir in the obtained AD–HN–Ir electrocatalyst with an ultralow content of approximately 3.5 μg cm$^{-2}$.

**Synthesis of the Ir–NC electrocatalyst.** The general process of synthesis is similar to that of the preparation of AD–HN–Ir electrocatalyst. It is noted that in the process of substrate surface functionalization, PAni was not treated with con-centrated ammonia water, and the final catalyst did not form heteronitrogen coordination and was called Ir–NC electrocatalyst. The mass loading of Ir of Ir–NC is approximately 3.0 μg cm$^{-2}$.

**Synthesis of Ir–NP/NC electrocatalyst.** The general process of synthesis is similar to that of the preparation of Ir–NC electrocatalyst. It is noted that in the process of the adsorption of the Ir precursor, it is necessary to increase the metal feeding amount to 0.18 mmol/L, and then, the carbonization temperature was increased to 1000 °C. Obvious Ir nanoparticles were obtained over 3D CP, which was called Ir–NP/NC electrocatalyst. The mass loading of Ir of Ir–NP/NC is approximately 8.5 μg cm$^{-2}$.

**Morphology and structure characterization.** TEM and high-resolution TEM (HRTEM) were carried out by a JEM-2100F microscope at an acceleration voltage of 200 kV. HAADF-TEM with aberration-corrected, conducted at 200 kV on a JEM-ARM200F instrument. SEM measurements were performed by an SEM (JSM-6700F, and 5 kV). X-ray photoelectron spectroscopy (XPS) spectra were acquired on an ESCALAB MKII, and the excitation source used Mg $K\alpha$ ($hv = 1253.6$ eV). The C and N $K$-edge XANES spectra were measured on the BL12B-a beamline of

the National Synchrotron Radiation Laboratory (NSRL, China). The measure mode uses total electron yield mode under a vacuum better than $5 \times 10^{-8}$ Pa.

**Electrochemical measurements**. All electrochemical measurements were performed in a three-electrode system by CHI760D electrochemical workstation, where the prepared electrodes immersed in a sulfuric acid electrolyte solution (0.5 M), $1 \times 1$ cm$^2$ CP with catalyst, carbon rod, and SCE act as the working, auxiliary, and reference electrode, respectively. The three electrodes were immersed in a sulfuric acid electrolyte solution (0.5 M) as conductive media at pH 0, which was placed in an N$_2$-purged flow to remove O$_2$ before measurement. Linear sweep polarization curves were measured in different temperatures (5, 15, 25, 35, and 45 °C) for acquiring activation energy based on Arrhenius plios ($k = A\exp(-E_a/RT)$, the chemical reaction rate constant ($k$), and thermodynamic temperature ($T$) and reaction activation energy $E_a$ and pre-exponential factor $A$ (also called frequency factor). Electrochemical data were corrected for uncompensated series resistance $R_s$, which was determined through the test under the open-circuit voltage, and then 90% IR compensation was selected to obtain $R_s$. The final polarization curve was obtained by IR compensation. The value of $R_s$ was 2.6 Ω in 0.5 M H$_2$SO$_4$. The potential was determined by the following Eq. (1):

$$E_{\text{Corrected}} = E_{\text{Uncorrected}} - I * R_s, \qquad (1)$$

where $I$ is the current.

**Calculation of the roughness factor (RF)**

The roughness factor (RF) is calculated by taking the estimated ECSA and dividing by the geometric area of the electrode, 2.0 cm$^2$, according to the following Eq. (2):

$$\text{RF} = \text{ECSA}/S_{\text{geo}}. \qquad (2)$$

The ECSA of the catalyst sample is calculated from the double layer capacitance according to the following Eq. (3):

$$\text{ECSA} = C_{dl}/C_s, \qquad (3)$$

where $C_s$ is the specific capacitance of the sample, which has been measured for a variety of electrodes in acidic solution, and the typical value reported ranges between $C_s = 0.015 - 0.110$ mF cm$^{-2}$ in H$_2$SO$_4$. Hence, we use general specific capacitances of $C_s = 0.035$ mF cm$^{-2}$ based on typical reported values. The double-layer capacitance ($C_{dl}$) and the ECSA can be calculated by CV tests. It is necessary to select measuring windows larger than 100 mV not only to ensure reaching a steady charging current but also to ensure that the $i_C$ values extracted from the anodic and cathodic scans are similar. Furthermore, it is recommended to use a scan rate range as wide as possible to reach suitable current values. Therefore, we performed CV tests at a wide redox process-free window of 300 mV and suitable sweep speeds of 0.1, 0.25, 0.50, 0.75, and 1.00 V/s[34]. This range is typically a 0.3 V potential window (0.7–1.0 V vs. SCE) centered at the open-circuit potential of the system.

Another activity metric sometimes reported in the electrocatalysis literature is the specific activity at a given overpotential. The definition of specific activity refers to the specific current per ECSA ($J_{\text{ECSA}}$), which is calculated by dividing the current density per geometric area ($J_{\text{geo}}$) by the RF at a given overpotential, according to the following Eq. (4):

$$J_{\text{ECSA}} = J_{\text{geo}}/\text{RF}. \qquad (4)$$

Turnover frequency (TOF) and mass activity calculations.

$$\text{TOF} = \frac{\text{the measured current density } j * NA * 10^3}{4 * F * \text{surface active sites (mol)}}, \qquad (5)$$

$F$ represents the Faraday constant (96,485 c cm$^{-1}$). $NA$ represents the Avogadro constant ($6.02 \times 10^{23}$). We estimated the number of active sites as the total number of surface sites of Ir for the electrocatalysts, which might underestimate the real TOF.

Mass activity (A g$_{\text{metal}}^{-1}$) values were calculated from the electrocatalyst loading $m$ and the measured current density $j$ at overpotentials of 216 and 260 mV

$$\text{Mass activity} = \frac{j}{m}. \qquad (6)$$

The S-number is defined as the ratio between the amount of evolved oxygen (calculated from $Q_{\text{total}}$) and the amount of dissolved iridium (extracted from ICP–MS data). Lifetimes of electrocatalyst was calculated on the basis of the equation[28]:

$$t = \frac{S \times z \times F \times m}{j \times M}, \qquad (7)$$

where $t$ is the lifetime of the electrocatalyst (s), $S$ is the stability number, $z$ is the number of electrons per evolved O$_2$, $F$ is the Faraday constant (96,485 C mol$^{-1}$), m is the loaded mass of iridium (g cm$^{-2}$), $j$ is the applied current density (A cm$^{-2}$) and $M$ is the molar mass of iridium (192.2 g mol$^{-1}$).

**In situ EIS measurements**. In situ characterization tandem electrochemical operation was carried out at the specified potential for 10 min to obtain the surface

chemical composition and structural information of the materials. EIS tests were performed at different potentials in the frequency range of 0.01–100,000 Hz.

**In situ SRIR measurements**. In situ SRIR measurements were performed at the infrared beamline BL01B of the National Synchrotron Radiation Laboratory (NSRL, China) by using a homemade top-plate cell, in which the infrared transmission window of the cell was made of ZnSe crystals with low light absorption (cut-off energy of ~625 cm$^{-1}$). The SRIR testing device consists of two carefully designed parts, including an IR spectrometer (Bruker 66 v/s) with a KBr beam splitter and various detectors (herein, a liquid-nitrogen-cooled MCT detector was used) and an IR microscope (Bruker Hyperion 3000) with a 16× objective, which can provide infrared spectroscopy measurements over a broad range of 15–4000 cm$^{-1}$ and a high spectral resolution of 0.25 cm$^{-1}$. Attentively, during the in situ SRIR measurements, the infrared signal is very sensitive to water molecules and is easily disturbed. By pressing the electrode and window to control the thickness of the water film on the micron scale, the infrared signal-to-noise ratio (SNR) is improved. To obtain information on the electrocatalyst surface and further improve the signal quality of SRIR spectra under working conditions, the reflection mode of infrared light with vertical incidence is adopted in the test. Each high-resolution infrared absorption spectrum with a resolution of 2 cm$^{-1}$ was obtained by averaging 514 scans. To avoid the signal difference caused by the sample falling off, a constant potential was applied to the electrocatalyst electrode for 20 min, and then all infrared spectral acquisitions were carried out. The background spectrum of the electrocatalyst electrode was acquired at an open-circuit voltage before each systemic OER measurement, and the measured potentials ranges of the OER were 1.0–1.45 V. Notably, to show a higher SNR of the in situ SRIR data, the curves were smoothed through 10 points of data processing, meaning that 10 consecutive points were averaged to improve the SNR of the curves.

**In situ XAFS measurements**. The in situ XAFS measurements of Ir $L_3$-edge were carried out at the 1W1B station in the Beijing Synchrotron Radiation Facility (BSRF), China. The storage ring of BSRF was operated at 2.5 GeV with a maximum current of 250 mA. The beam from the bending magnet was monochromatized utilizing a Si (111) double-crystal monochromator and further detuning of 15% to remove higher harmonics. The electrochemical in situ XAFS tests were performed by a homemade cell in a 0.5 M H$_2$SO$_4$ electrolyte. The XAFS spectra were collected through fluorescence mode with a 19-element solid-state detector. The AD–HN–Ir electrocatalyst on the hetero-nitrogen-configured 3D carbon substrate was cut into $1 \times 2$ cm$^2$ pieces and then sealed in a cell by Kapton film. To obtain the evolution information of the active site during the electrochemical reaction, a series of representative potentials (1.25–1.45 V) were applied to the electrode. During the collection of XAFS measurements, the position of the absorption edge ($E_0$) was calibrated using a standard sample of Ir, and all XAFS data were collected during one period of beam time.

**XAFS data analysis**. The acquired EXAFS data were processed according to the standard procedures using the ATHENA module implemented in the IFEFFIT software packages. Subsequently, $k^2$-weighted $\chi(k)$ data in the k-space ranging from 3.0–11.8 Å$^{-1}$ were Fourier transformed to real space using a Hanning window (d$k$ = 1.0 Å$^{-1}$) to separate the EXAFS contributions from different coordination shells. The best background removal was at the $R_{bkg} = 1.1$ Å, and the low-frequency noise was removed fully. The amplitude reduction factor $S_0^2$ (0.78) was obtained from the $L_3$-edge EXAFS curve fitting for phthalocyanine iridium. As for the ex situ for AD–HN–Ir electrocatalyst, the curve fitting was done on the $k^2$-weighted EXAFS function $\chi(k)$ data in the $k$-range of 3.0–11.8 Å$^{-1}$ and in the $R$-range of 1.0–2.5 Å. The number of independent points for these samples are $N_{ipt} = 2\Delta k \cdot \Delta R/\pi = 2 \times (11.8 - 3.0) \times (2.5 - 1.0)/\pi = 8$. However, compared with that of the ex-situ sample, the first coordination peaks of in situ AD–HN–Ir electrocatalysts under 1.25, 1.35, and 1.45 V show an obviously increased strength and a higher $R$ shift, which was ascribed to the addition of Ir–O coordination. Therefore, the two subshells of Ir–N and Ir–O coordination were considered for the curve fitting of AD–HN–Ir electrocatalysts under work conditions. During curve-fittings, each of the Debye–Waller factors ($\sigma^2$), coordination numbers ($N$), interatomic distances ($R$), and energy shift ($\Delta E_0$) was treated as adjustable parameters. Notably, in order to reduce the number of adjustable fitting parameters, the $N$ and $\Delta E_0$ of Ir–N coordination shell were fixed and equal to the parameters of the ex situ sample. Quantitative curve fittings were carried out for the Fourier transformed $k^2\chi(k)$ in R-space using the ARTEMIS module of IFEFFIT. The amplitude $F_i (k, r_i)$ and phase shifts $\Phi_i(k, R_i)$ function for the Ir-N/O atom pairs were calculated by the FEFF6 in the muffin-tin (MT) self-consistent-field approximation. Default values of MT radii and MT radii overlap of 15% between contiguous spheres were used to simulate the atomic bond. To account for energy-dependent exchange-correlation potential, the energy- and position-dependent optical Hedin–Lundqvist potential was employed. The calculations were performed for a cluster with a radius of 8 Å having the structure of phthalocyanine iridium (iridium oxide) and centered at the Ir–N (Ir–O) atom, respectively.

## Data availability

All the data reported in this paper are available from the corresponding author upon request.

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

## Acknowledgements

This work was supported by the National Key Research and Development Program of China (2017YFA0402800), the National Natural Science Foundation of China (Grants nos. U1932212, U1932109, 12135012, 11875257, and 12111530002), the Postdoctoral Science Foundation of China (2020M682042).

## Author contributions

S.Q.W., Q.H.L., and H.S. conceived the project. Q.H.L. and H.S. designed the in situ SRIR and XAFS experiments. H.S., W.L.Z., and W.Z. carried out the experiments. S.Q.W., Z.M.Q., L.R.Z., H.Z., Y.L.L., M.H.L., X.X.Z., X.S., Y.Z.X., F.C.H., J.Z., and T.D.H. analyzed the experimental data. The paper was written by S.Q.W., Q.H.L., and H.S. with contributions from all authors.

## Competing interests

The authors declare no competing interests.
