## [Peer Review File · Nature Communications]

REVIEWER COMMENTS

Reviewer #1 (Remarks to the Author):

The manuscript by Su et al. is addressing an interesting topic related to the electrocatalytic oxygen evolution reaction (OER). The choice of methods is appropriate, and the authors performed controlled experiments such as operando X-ray absorption fine structure and infrared spectroscopy under OER working conditions for investigating the catalytic dynamic process. The results are likely to be of wide interest to catalytic reaction mechanisms and catalyst design. Overall, I consider that both the significance and the novelty of this work could satisfy the high standards of Nature Communications. Therefore, I recommend the publication of this manuscript after the following issues are clarified:

1. The design principle of the catalysts should be further clarified. In the experimental step, why did the authors select nitric acid rather than hydrochloric acid or sulfuric acid during the electrodeposition process?
2. Temperature is an important factor in the synthesis process of the samples in this work. The temperature of carbonization will affect the structure of carbon and then affect the performance of the sample. Did the author try other temperatures?
3. The author mentioned that an oxygen is dynamically coupled at the Ir site under the driven voltage, which then promotes the formation of the key *OOH intermediate. In the present manuscript, however, it is not clear enough how to determine the relationship between the oxygen coupling and the adsorption of oxygen-containing intermediates. Some related discussions should be strengthened through the manuscript.
4. Environment for electrochemical testing has a certain effect on the performance of the catalyst. The potential temperature effects should be considered and the related discussion on this issue should be included.
5. The presentation of the experimental results in the manuscript should be further improved. For example, some of the potentials listed throughout the manuscript need to be listed/provided relative to a reference electrode.

Reviewer #2 (Remarks to the Author):

In this manuscript, single-atomic iridium based ultralow-iridium electrocatalyst was synthesized via "electric-driven amino-induced" strategy, which delivers a low overpotential of 216 mV to achieve a current density of 10 mA cm⁻². With the operando synchrotron radiation X-ray absorption fine structure (XAFS) spectroscopy analysis, the author revealed that oxygen was dynamically pre-adsorbed on the Ir site in the form of O-hetero-Ir-N₄ moiety under low driven-potential to accelerate the transfer of electrons from the metal sites to neighboring atoms. The work is scientifically sound and well presented. However, some questions remain unresolved regarding the origin of the observed stability of the catalyst. The stability of catalysts is a more important issue than the catalytic activity for the practical application. A suitable explanation for the observed stability of the catalyst would open a great avenue for further developments of single atom catalysts. With that part augmented, this work would be a welcome addition to Nature communications.

Reviewer #3 (Remarks to the Author):

Su et al. report well dispersed Ir on N-doped carbon derived from polyaniline (PANI) as an electrocatalysts for the oxygen evolution reaction in acid. The material and its activity are characterized by SEM, TEM, STEM_EDS, XAS at the C-K, N-K, Ir-L3 edges, XPS, synchrotron IR spectroscopy, cyclic voltammetry and impedance spectroscopy. Ir is a known active site for the OER in acid.

I see the main achievements of the authors in reducing its loading while maintain high activity and stability, which is on the roadmap of many international initiatives. The authors also convincingly explain why their catalyst is more active than simply putting Ir on PANI.

The main claims are

C1 – Large mass activity

C2 – High stability

C3 – The good performance is due to pre-adsorption and facile OOH production

C1. I checked this carefully and I agree that the produced electrodes are among the highest reported for the OER in acid. I do not find that data/references in Supplemental Table 2 are sufficient to support the claim.

Please cite and include the state-of-the-art catalysts from recent reviews, e.g.

<https://onlinelibrary.wiley.com/doi/full/10.1002/adma.201806296>

<https://pubs.rsc.org/en/content/articlelanding/2020/nr/d0nr02410d#!divAbstract>

A previous report in Nat. Commun should be discussed and compared in detail:

<https://www.nature.com/articles/s41467-019-12886-z>

What are the major steps forward in this work? (and why is this not included in the Suppl. Table?)

How reproducible are the activity results? How many electrodes were prepared and tested? Please add error bars to support a consistently large mass activity.

C2 - The electrodes have high stability at 10 mA/cm². However, this is not a realistic condition for a PEM electrolyzer. I appreciate that at least the activity was given at 100 mA/cm², which also is too low to be useful for an electrolyzer.

Does the catalyst electrode survive 2000 mA/cm² and 2 V cell potential, which are current state-of-the-art performance parameters for PEM electrolyzers, e.g. outlined in https://www.energy-x.eu/wp-content/uploads/2020/02/Energy_X_Research-needs-report_final_24.02.2020.pdf

How long do the electrodes last under these more realistic high current/voltage conditions?

Is it possible for the authors to estimate the lifetime under PEM electrolyzer conditions using the S-number (https://www.nature.com/articles/s41929-018-0085-6?WT.feed_name=subjects_hydrogen-energy), e.g. by measurement of Ir loss to the electrolyte?

C3 – I find the explanation and data plausible to explain the increase in performance. Yet, the discussion seems to rely on results that are presented later in the text. I comment on it below and recommend to move the discussion after the presentation of all data/figures.

On balance, I find the study potentially suitable for Nature Communications if the claims were better supported as suggested and if there is sufficient novelty relative to a similar publication in this journal (<https://www.nature.com/articles/s41467-019-12886-z>).

Please also address the following points:

1) Title

a) The title states "single atom", yet Fig. 1c is too small to judge and Fig. S5 shows larger spots (near the 5 of the scale bar). How many spots were analyzed? Is the selection of Fig. S5b representative? I find the expression "atomically dispersed Ir" much better than claiming all single atoms, which may not be supported by EXAFS analysis, see below.

b) The study is in situ but not operando. Operando requires a simultaneous product measurement (e.g. <https://www.sciencedirect.com/science/article/pii/S0920586104008065?via=ihub>). That could be the current for electrocatalysis but no data from in situ experiments is shown. Please either add this data (which must be convincing) or use "in situ".

c) I am not sure what is meant by "dynamic-coupling oxygen". Oxygen species adsorb and desorb during the catalytic cycle. This is expected. In order to oxidize hydroxide/water, one must adsorb it and then release O₂. I recommend to remove the confusing expression or discuss in detail what

new information it adds. Also in the main text, the word "dynamic" seems to have no important meaning and most sentences would contain the same information when it was deleted.

2) Please add the source of the Ir atoms to the main text. It was hard to find this important information in the

3) EXAFS analysis

a) Please add the details of the Fourier transform and used window function, otherwise the figure is not reproducible.

b) Please include the data in k-space, which contains more information

c) P7. What is meant by "oscillation frequency"? The EXAFS oscillations in Fig. 2e? Only one period is shown and it matches that of IrO₂ and not that of a material with Ir-N bonds

d) How were the fits performed? In k-space in which range or in R-space or Fourier-filtered k-space (Q-space in Artemis)? This is important to reproduce the work and judge its meaningfulness

e) What structural model was used and what were the parameters to obtain the phase functions?

f) The red line in Fig. 2f has peaks at higher R. Can they be fit with Ir-Ir interactions (thus indicating no single atom Ir)? why should the peaks in the FT around 3 Å not be significant? They look well above the noise level and fall into the range expected for Ir-Ir distances.

g) Is a better background subtraction possible for the data in Fig. 5b? Can it be excluded that the difference in the analyzed peak height is due to the distortion that causes the peak at 1 Å? The k-space data would help experts to identify issues with the data and judge if the analysis of the peak at 1.6 Å is sound.

4) Activity

a) Please test the activity and stability also at higher current densities such as 500 mA/cm², 1000 mA/cm² and 2000 mA/cm² if possible. PEM electrolyzers are operated under harsher conditions and I am not convinced stability at 10 mA/cm² is relevant for this application.

b) Please repeat at least the activity measurements to get statistics.

c) It is good to perform measurements in a two-electrode cell but the one used by the authors is by no means close enough to a PEM electrolyzer system make statements about "practical electrolyzers"

d) P15. 100VmA is not a universally suitable potential window for Cdl determination by cyclic voltammetry and the range of sweep speeds is rather narrow

(<https://iopscience.iop.org/article/10.1088/2515-7655/abee33>). This can introduce large systematic errors in the determination of specific activity (in Fig. S11).

5) At what current density are the Tafel slopes evaluated? Please state explicitly. There seem to be at least two Tafel slopes in the plot of each trace and the traces are not very linear in this plot

6) It was not quite clear to me during the first read why the catalyst electrode is compared to PANI-Ir. Please revise the text with a statement that the catalyst Ir-NC is based on PANI-Ir but has an additional step in the preparation procedure.

7) P10. How can R_{ct} be used to represent the H₂O/OOH ion adsorption resistance? R_{ct} relates to charge transfer across an interface. Does the statement make use of the SRIR results discussed later? The connection between R_{ct} and "adsorption kinetics" is likewise unclear. Please revise. In the first half of P10, there is no data yet that OOH is created at all.

8) P14. Conclusion: "Hence, this discovery of active sites evolution under working conditions can provide a coordination-engineered strategy for designing advanced acidic-OER electrocatalysts." What is meant? How exactly are the reported findings useful for other electrocatalysts? What should other scientists look for in good OER catalysts in acid? Finally, what is meant by "active sites evolution"? Species must adsorb at the active site? Does this refer to the O adsorption on the other side? In the latter case, does it desorb again at lower potential or stay there? The insight that the active state of electrocatalysts differs from the as-made material is not novel and was reported numerous times.

7) The panel labels a,b,c,... seem to be missing for most of the figures.

8) P17. How was the data smoothed? What method? What was the difference between neighboring steps? 2 cm⁻¹? 100 points could smooth the spectra too much. Please also show the unprocessed data.

9) Please have the manuscript checked by a native English speaker. It is largely understandable but some sentences do not read well (e.g. The N K-edge in Figure 2b, the peaks of excitation of C-N and hetero-Ir-N were obtained.)

Report by Marcel Risch

Response to Reviewers' Comments

We are grateful to the reviewers for having given us important and valuable comments on the manuscript NCOMMS-21-20746 entitled:

“*In-situ* spectroscopic observation of dynamic-coupling oxygen on atomically-dispersed iridium electrocatalyst for acidic water oxidation”

by Hui Su, Wanlin Zhou, Wu Zhou, Yuanli Li, Lirong Zheng, Hui Zhang, Meihuan Liu, Xiuxiu Zhang, Xuan Sun, Yanzhi Xu, Fengchun Hu, Jing Zhang, Tiandou Hu, Qinghua Liu, * and Shiqiang Wei*

We have read your comments and criticisms seriously. At first, we greatly appreciate the reviewers for your high evaluations on this manuscript: “*The results are likely to be of wide interest to catalytic reaction mechanisms and catalyst design. Overall, I consider that both the significance and the novelty of this work could satisfy the high standards of Nature Communications* (Reviewer #1)”, “*The work is scientifically sound and well presented* (Reviewer #2).”, and “*I see the main achievements of the authors in reducing its loading while maintain high activity and stability, which is on the roadmap of many international initiatives* (Reviewer #3)”. Meanwhile, the reviewers have raised numerous constructive comments on the stability results and the spectroscopy analysis, which are quite helpful for further improving the quality of this work. According to the reviewers’ constructive suggestions, we have performed necessary supplemented experiments and detailed analysis, and obtained significant new results to substantially support the hypotheses and conclusions of this work. Accordingly, we have seriously modified this manuscript and we hope that the revised manuscript could meet the high standard of *Nature Communications*. The detailed replies to your comments are presented in a point-to-point manner as follows.

REVIEWER COMMENTS

Reviewer #1 (Remarks to the Author):

The manuscript by Su et al. is addressing an interesting topic related to the electrocatalytic oxygen evolution reaction (OER). The choice of methods is appropriate, and the authors performed controlled experiments such as operando X-ray absorption fine structure and infrared spectroscopy under OER working conditions for investigating the catalytic dynamic process. The results are likely to be of wide interest to catalytic reaction mechanisms and catalyst design. Overall, I consider that both the significance and the novelty of this work could satisfy the high standards of Nature Communications. Therefore, I recommend the publication of this manuscript after the following issues are clarified:

1. Question: The design principle of the catalysts should be further clarified. In the experimental step, why did the authors select nitric acid rather than hydrochloric acid or sulfuric acid during the electrodeposition process?

Reply: This is a very good question. We are sorry that the principles for selecting electrolyte in electrodeposition were not clearly described in the previous manuscript. Nitric acid was chosen as the electrolyte mainly for the exclusion of impurities and it would be discussed in more detail below.

In this work, in order to obtain atomically dispersed Ir atoms with covalent coupling in a hetero-nitrogen-configured 3D carbon substrate, the most key step is first electrodeposition of polyaniline layer onto a 3D carbon substrate and then the surface was functionalized with the amino group for strongly bonding to Ir complex ions. Firstly, in the electrodeposition process, the most important point is to avoid introducing heteroatoms in the electrodeposition process of polyaniline. The Cl (HCl) and S (H₂SO₄) are easily introduced into the polyaniline skeleton to form the doping of Cl and S elements in the later high temperature annealing. Therefore, nitric acid, which contains no miscellaneous elements, is the best choice for acidic electrolytes in electrodeposited polyaniline. Furthermore, the NO₃⁻ is easily decomposed at high temperature without introducing any miscellaneous elements for catalysts. Above all, considering the

avoidance of impurity doping of anions, we chose HNO₃ as electrolyte in the electrodeposition process.

Accordingly, in line 21, page 16 of the revised manuscript, the following text **has been added**: “*Considering the avoidance of impurity doping of anions, we chose HNO₃ as electrolyte in the electrodeposition process.*”

2. Question: Temperature is an important factor in the synthesis process of the samples in this work. The temperature of carbonization will affect the structure of carbon and then affect the performance of the sample. Did the author try other temperatures?

Reply: Thank you for your careful consideration. Specifically, the polyaniline electrode is heated to 800 °C for 3 h to obtain the AD-HN-Ir electrocatalyst. At 800 °C, polyaniline would produce pyrolysis and be gradually carbonized to porous highly conductive hetero-nitrogen-configured 3D carbon materials. At the same time, the Ir atoms were chemically coupled with the N sites to form the hetero-Ir-N₄ active centers on a hetero-nitrogen-configured 3D carbon substrate.

Noticeably, the graphitization of the substrate has a great influence on the properties, and the graphitization trend of the carbon substrate will be more obvious with the increase of temperature. However, the disadvantages of higher temperature are the collapse of substrate morphology and the agglomeration of single atoms on the surface, thus reducing the catalytic activity (*Adv. Energy Mater.* 8, 1702476 (2018)). Considering that the pyrolysis temperature will affect the carbonization degree of polymer and the distribution of Ir, the samples were also performed in different pyrolysis temperatures (700 °C and 900 °C). As shown in the Figure N1, the as-synthesized Ir-700 samples (pyrolysis temperature of 700 °C) has clear dendritic structure similar to polyaniline and delivers an overpotential of 285 mV at 10 mA cm⁻², which reveals low electron conduction and material diffusion attributed to the inadequate polyaniline carbonization at a lower temperature. Furthermore, Figure N1d clearly shows that the small Ir nanoparticles were distributed on the 3D carbon substrate and a higher overpotential of 252 mV at 10 mA cm⁻² was observed for the as-

synthesized Ir-900 sample (pyrolysis temperature of 900 °C), suggesting Ir atoms aggregated into nanoparticles at a higher pyrolysis temperature. Above all, considering the conductivity of the carbon substrate and the phase composition of Ir sites, the pyrolysis temperature of 800 °C was selected for AD-HN-Ir electrocatalyst. It can insure a stable hetero-Ir-N₄ moieties spatially confined in the hetero-nitrogen-configured 3D carbon substrate for an efficient 4-electron OER process.

Figure N1. (a) Linear sweep voltammetry (LSV) curves for Ir electrocatalysts annealed at 700, 800 and 900 °C. SEM images of Ir-based electrocatalysts annealed at (b) 700 °C, (c) 800 °C and (d) 900 °C.

Accordingly, the Figure N1 has been added in the Supplementary Information as Supplementary Fig. 10. In line 18, page 8 of the revised manuscript, the following text **has been added:** “*The morphology structure and OER performance of Ir electrocatalysts at different pyrolysis temperatures were shown in Supplementary Fig. 10, revealing the AD-HN-Ir (800 °C) electrocatalyst with better carbonization degree and atomically dispersed Ir active sites.*”

3. Questions: The author mentioned that an oxygen is dynamically coupled at the Ir

site under the driven voltage, which then promotes the formation of the key *OOH intermediate. In the present manuscript, however, it is not clear enough how to determine the relationship between the oxygen coupling and the adsorption of oxygen-containing intermediates. Some related discussions should be strengthened through the manuscript.

Reply: We are greatly grateful to the reviewer for your nice question and it is useful for improving the quality of this work. The atomically dispersed Ir active sites coupled onto hetero-nitrogen-configured 3D carbon substrate can form stable and highly active O-hetero-Ir-N₄ structure, obviously accelerating the 4-electron kinetics process and promoting the formation of key OER reaction intermediates *OOH for efficient acidic-OER activity.

Firstly, a new absorption band at 784 cm⁻¹ was observed by *in-situ* SRIR measurements for AD-HN-Ir electrocatalyst at potential of 1.25 V, which can be assigned to the emergence of Ir-O during OER process. Simultaneously, the *in-situ* XAFS results reveal that the intensity of white-line peak of AD-HN-Ir electrocatalyst increases gradually with the increase of voltage, accompanied by a slight positive-shift, indicating more electrons move from Ir to nearby atoms and oxygen-species. Interestingly, the peak intensity increased significantly at the potential of 1.25 V and the fitting result clearly shows an additional first shell of Ir-O coordination, suggesting a local structural self-optimization and adsorption of reactive species under OER working conditions, which optimizes the electronic structure of the active site and speeds up the production of the reaction intermediate. Importantly, a new absorption band in the vibration frequency of 1055 cm⁻¹ was observed for AD-HN-Ir electrocatalyst when the potential > 1.35 V, which could be attributed to the emergence of crucial intermediate *O-OH during the acidic-OER process. Furthermore, the key superoxide intermediate formed at the active site was further confirmed by the emergence of new first shell of Ir-O coordination in the *in-situ* XAFS results. Above all, high-valence Ir active sites coupled onto hetero-nitrogen-configured 3D carbon substrate by strong coupling substrate effect can dynamically couple one oxygen atom

to form stable and highly active O-hetero-Ir-N₄ structure, obviously accelerating the 4-electron kinetics process and promoting the formation of key OER reaction intermediates OOH for efficient acidic-OER activity.

Accordingly, in line 20, page 15 of the revised manuscript, the following text **has been added**: “Above all, the in-situ SRIR and XAFS results jointly reveal that Ir active sites coupled onto the hetero-nitrogen-configured 3D carbon substrate by a strong coupling substrate effect can dynamically couple one oxygen atom to form a stable and highly active O-hetero-Ir-N₄ structure during the reaction process, obviously accelerating the 4-electron kinetics process and promoting the formation of key OER reaction intermediates OOH for efficient acidic-OER activity (Figure 5d).”

4. Question: Environment for electrochemical testing has a certain effect on the performance of the catalyst. The potential temperature effects should be considered and the related discussion on this issue should be included.

Reply: Thank you for your nice questions. We really praise the reviewer for your expert knowledge in the field of electrochemistry. According to your suggestion, the electrochemical measurements were performed under different temperatures to seek the relationship between working temperature and activity. Seen from Figure N2, it can be drawn that the AD-HN-Ir electrocatalyst delivers faster kinetics and better activity at a higher temperature.

The temperature of the electrochemical cell is an important parameter that significantly affects the electrochemical performance tests (*J. Am. Chem. Soc.* 140, 2926–2932 (2018)). Generally, the performance tests of the electrochemical cell were carried out at room temperature. In order to further explore the temperature effect, the electrochemical cell was put on the temperature-control heater with temperature controlled at 25, 50 and 80 °C for temperature-dependent studies. As shown in the Figure N2, the OER activity are quite temperature-dependent, and the overpotentials of AD-HN-Ir electrocatalyst decrease from 216 to 198 mV at a current density of 10 mA cm⁻² with the increase of temperature to 80 °C, which is consistent with what has been

observed that a higher temperature provides faster OER kinetics and better OER activity (*J. Power Sources* 167, 235–242 (2007)). Based on the above results, the AD-HN-Ir electrocatalyst shows temperature-dependent OER activity with faster OER kinetics at higher temperatures.

Figure N2. Linear sweep voltammetry (LSV) curves of AD-HN-Ir under 25, 50 and 80 °C in 0.5 M H₂SO₄.

Accordingly, Figure N2 has been added in the Supplementary Information as Supplementary Fig. 11. In reverse line 2, page 8 of the revised manuscript, the following text **has been added**: “*Considering the effect of temperature on acidic OER performance, we conducted a water splitting test under different operating temperatures. The OER activity is quite temperature-dependent, and the overpotentials of AD-HN-Ir electrocatalyst decrease from 216 to 198 mV at a current density of 10 mA cm⁻² with the increase of temperature to 80 °C, which is consistent with what has been observed that a higher temperature provides faster OER kinetics and better OER activity (Supplementary Fig. 11).*”

5. Question: The presentation of the experimental results in the manuscript should be further improved. For example, some of the potentials listed throughout the manuscript need to be listed/provided relative to a reference electrode.

Reply: This is a good suggestion to improve the readability of the manuscript. According to your suggestion, we have carefully checked the entire manuscript, and the voltages in the manuscript are all relative to reversible hydrogen electrode, and the potentials are relative to saturated calomel electrode in the experimental preparation.

Reviewer #2 (Remarks to the Author):

In this manuscript, single-atomic iridium based ultralow-iridium electrocatalyst was synthesized via “electric-driven amino-induced” strategy, which delivers a low overpotential of 216 mV to achieve a current density of 10 mA cm⁻². With the operando synchrotron radiation X-ray absorption fine structure (XAFS) spectroscopy analysis, the author revealed that oxygen was dynamically pre-adsorbed on the Ir site in the form of O-hetero-Ir-N₄ moiety under low driven-potential to accelerate the transfer of electrons from the metal sites to neighboring atoms. The work is scientifically sound and well presented. However, some questions remain unresolved regarding the origin of the observed stability of the catalyst. The stability of catalysts is a more important issue than the catalytic activity for the practical application. A suitable explanation for the observed stability of the catalyst would open a great avenue for further developments of single atom catalysts. With that part augmented, this work would be a welcome addition to Nature communications.

Reply: We are greatly grateful to the reviewer for your nice question and it is useful for improving the quality of this work. According to your suggestion, the stability of acidic-OER for AD-HN-Ir electrocatalyst will be analyzed in detail from three aspects: preparation, structure and performance.

The operation stability is an important index to evaluate the performance of electrocatalyst, especially for acidic OER. It is known that for highly reactive iridium electrocatalysts, especially for commercial IrO₂, the high oxidation voltage causes the excessive increase of oxidation state and the decrease of crystallization of the electrocatalyst (*J. Am. Chem. Soc.* 139, 12837–12846 (2017); *J. Electroanal. Chem.* 774, 102–110 (2016)). Additionally, a number of recent studies have unveiled the *in-situ* reconstruction of electrocatalyst surface, which can in turns affect its intrinsic OER-activity due to a significant change of the surface oxidation state as well as agglomeration and dissolution of surface active phases (*Nat. Catal.* 1, 300–305 (2018); *Adv. Mater.* 30, 1804333 (2018)). To improve the durability of the electrocatalyst under acidic OER operation condition, the key is to construct stable coordination structure of

Ir active sites to inhibit the peroxidation and the aggregation and dissolution of the surface active phase under high potentials.

Preparation: In this work, the AD-HN-Ir electrocatalyst with atomically dispersed Ir active sites coupling in the hetero-nitrogen-configured 3D carbon substrate was synthesized via a controllable “electric-driven amino-induced” strategy. The polyaniline layer was first electrodeposited onto a 3D carbon substrate, and then functionalized with amino groups by mild heat treatment in ammonia solution. It is noted that the surface reductive benzenoid-amine groups and -NH₂-derived uncoordinated N sites act as the anchoring sites for Ir atoms during ions exchange and pyrolysis processes, and the atomically dispersed Ir active sites with stable hetero-Ir-N₄ moieties were strongly coupled on the 3D carbon substrate. This stable configuration has the potential to inhibit the peroxidation and aggregation of Ir active sites at high potentials.

Structure: The *in-situ* XAFS and SRIR techniques reveal that one oxygen atom was dynamically coupled on the Ir active site in the form of O-hetero-Ir-N₄ moiety under low driven-potential, and then H₂O adsorption under a low driven-potential and crucial *OOH intermediate production were achieved. Simultaneously, the *in-situ* XAFS results under working conditions doubtlessly proved that the Ir active sites were only weakly oxidized to a highly active oxidation state and remains atomically dispersed after long-time operation. Most importantly, one oxygen atom was dynamically formed on the Ir active site with stable O-hetero-Ir-N₄, which endows higher coordination numbers and higher 5d electron occupied states at the Ir active sites to inhibit the dissolution of Ir sites for high activity and durability.

Performance: The stability tests at potentials of 1.53 and 1.75 V were performed. The activity of acidic OER did not decrease significantly after continuous operation for 50 h under a low current density of 10 mA cm⁻² (Figure 3f) and a high current density of 100 mA cm⁻² (Figure N3). Meanwhile, the morphology characterization showed that no collapse of the carbon substrate and no obvious particles were observed. Above all, the AD-HN-Ir electrocatalyst with atomically dispersed hetero-Ir-N₄ moieties delivers

high durability due to the optimization of electronic and coordination structure by the dynamic oxygen formed over the Ir sites.

Figure N3. (a) OER stability for AD-HN-Ir electrocatalyst at the applied potential of 1.53 V, and (b) SEM image of AD-HN-Ir electrocatalyst after reaction.

Accordingly, the Figure N3 has been added in Supplementary Information as Supplementary Fig. 21a and b. In line 17, page 10 of the revised manuscript, the following text **has been added:** “*In summary, the high durability of AD-HN-Ir electrocatalyst under continuous operating conditions is attributed to the stable hetero-Ir-N₄ moieties, which can inhibit peroxidation and dissolution of Ir active sites.*”

Reviewer #3 (Remarks to the Author):

Su et al. report well dispersed Ir on N-doped carbon derived from polyaniline (PANI) as an electrocatalysts for the oxygen evolution reaction in acid. The material and its activity are characterized by SEM, TEM, STEM_EDS, XAS at the C-K, N-K, Ir-L3 edges, XPS, synchrotron IR spectroscopy, cyclic voltammetry and impedance spectroscopy. Ir is a known active site for the OER in acid.

I see the main achievements of the authors in reducing its loading while maintain high activity and stability, which is on the roadmap of many international initiatives. The authors also convincingly explain why their catalyst is more active than simply putting Ir on PANI.

The main claims are

C1 – Large mass activity

C2 – High stability

C3 – The good performance is due to pre-adsorption and facile OOH production

Reply: Thank you for your positive evaluation on our work. The specific responses to your constructive questions are as follows.

C1. I checked this carefully and I agree that the produced electrodes are among the highest reported for the OER in acid. I do not find that data/references in Supplemental Table 2 are sufficient to support the claim.

Please cite and include the state-of-the-art catalysts from recent reviews, e.g.

<https://onlinelibrary.wiley.com/doi/full/10.1002/adma.201806296>

<https://pubs.rsc.org/en/content/articlelanding/2020/nr/d0nr02410d#!divAbstract>

A previous report in Nat. Commun should be discussed and compared in detail:

<https://www.nature.com/articles/s41467-019-12886-z>

What are the major steps forward in this work? (and why is this not included in the Suppl. Table?)

Reply: We thank the reviewer for the constructive suggestion that is quite useful for improving the quality of this work. According to your suggestion, more detailed

research results and related reference literature are rearranged and presented in Table N1. The AD-HN-Ir electrocatalyst showed optimal electrocatalytic activity according to the mass activity results at a lower overpotential of 216 mV.

Mass activity is an important parameter for the evaluation of the intrinsic activity of electrocatalysts. As seen from Table N1, the mass activity of the AD-HN-Ir electrocatalyst is up to $2860 \text{ A g}_{\text{metal}}^{-1}$ at the overpotential of 216 mV, which is remarkably larger than those of the reported electrocatalysts and the commercial IrO_2 . It is noted that a recent reported Ru-N-C catalyst also displays a high mass activity of $3571 \text{ A g}_{\text{metal}}^{-1}$ at an overpotential of 267 mV (Nat. Commun. 10, 4849 (2019)). For better comparison, the mass activity of the AD-HN-Ir electrocatalyst at the overpotential of 260 mV can reach $10900 \text{ A g}_{\text{metal}}^{-1}$, which is obviously better than those activities of other reported electrocatalysts.

Table N1. Comparison of the OER activity of the AD-HN-Ir electrocatalyst with other recently reported electrocatalysts in acid solution.

Catalysts	Mass loading (mg cm ⁻²)	Mass activity (A g _{metal} ⁻¹)	Overpotential at 10 mA cm ⁻² (mV)	References
AD-HN-Ir	3.5 μg _{Pt} cm ⁻²	2860(@216 mV)	216	This work
		10900(@260 mV)		
IrOx/ATO	0.01	39.1(@280 mV)	360	J. Am. Chem. Soc. 138 , 12552-12563 (2016).
IrNiCu DNF	0.16	460(@300 mV)	303	ACS nano 11 , 5500-5509 (2017).
6H-SrIrO ₃	0.9	75(@248 mV)	248	Nat. Commun. 9 , 5236 (2018)
Y ₂ [Ru _{1.6} Y _{0.4} O _{7.8}]	~0.06	~600(@270 mV)	250	Angew. Chem. Int. Ed. 130 , 14073-14077 (2018)
NaRuO ₂	0.2	42(@250 mV)	225	Adv. Energy Mater. 9 , 1803795 (2019)
Ru-N-C	0.28	3540(@267 mV)	267	Nat. Commun. 10 , 4849 (2019)
Ru ₁ -Pt ₃ Cu	0.016	779(@250 mV)	220	Nat. Catal. 2 , 304-313 (2019)
Cr _{0.6} Ru _{0.4} O ₂	0.28	229(@270 mV)	178	Nat. Commun. 10 , 162 (2019)
DO-IrNi _x	0.01	40(@250 mV)	--	Chem. Sci. , 5 , 2955 (2014)
La ₂ LiIrO ₆	0.016	47(@270 mV)	--	Nat. Energy 2 , 16189 (2016)
Ba ₂ YIrO ₆	0.015	--	350	Nat. Commun. 7 , 12363 (2016)
IrOx/SrIrO ₃	--	--	~270	Science 353 , 6303 (2016)
Li-IrOx	0.1	100(@290 mV)	270	J. Am. Chem. Soc. 141 , 3014(2019)
Np-Ir ₇₀ Ni ₁₅ Co ₁₅	--	--	220	Nano Energy 59 , 146 (2019)
Ir _{0.7} Ru _{0.3} O ₂	0.075	--	200	Adv. Energy Mater. 9 , 1802136 (2019)
Cu-Dope RuO ₂	0.192	52(@188 mV)	188	Adv. Mater. 30 , 1801351(2018)

Accordingly, the previous Supplementary Table 2 has been replaced by Table N1.

The reported electrocatalysts in published works (*Adv. Mater.* 31, 1806296 (2019), *Nanoscale* 12, 13249-13275 (2020) and *Nat. Commun.* 10, 4849 (2019)) have been included in the Table 2 in the revised Supplementary Information.

How reproducible are the activity results? How many electrodes were prepared and tested? Please add error bars to support a consistently large mass activity.

Reply: That is a good question. In the performance test, the electrode needs to undergo continuous CV test first, and then the realistic performance test is carried out after the electrode is stable to ensure the reliability of performance results. In addition, samples and electrodes were prepared repeatedly, and the results presented in the manuscript were all obtained after at least 10 tests of different batches of electrodes, which can ensure the repeatability of performance results. According to your suggestion, the error bars have been added in the Figure N4 to support their consistently large mass activities.

Figure N4. (a) Overpotentials at 10 and 100 mA cm⁻² for AD-HN-Ir, Ir-NC, IrO₂ and Ir-NP/NC electrocatalysts. (b) Mass activity (M.A.) and turnover frequency (TOF) at overpotentials of 216 and 260 mV for AD-HN-Ir and IrO₂ electrocatalysts.

Accordingly, the Figure 3b and 3e in the previous manuscript have been replaced by Figure N4a and 4b in revised manuscript. In line 18, page 9 of the revised manuscript, the following text **has been added**: “Notably, the mass activity and TOF of the AD-HN-Ir electrocatalyst are approximately 2860 A g_{metal}⁻¹ and 5110 h⁻¹ at a small overpotential of 216 mV, ~480 and 510 times larger than those of the commercial IrO₂ (6 A g_{metal}⁻¹, 10 h⁻¹) by performance test statistics.”

C2 - The electrodes have high stability at 10 mA/cm². However, this is not a realistic condition for a PEM electrolyzer. I appreciate that at least the activity was given at 100 mA/cm², which also is too low to be useful for an electrolyzer.

Reply: We thank the reviewer for this constructive suggestion. According to your suggestion, the stability at 100 mA cm⁻² was performed with the test results shown in Figure N5. The AD-HN-Ir electrocatalyst reaches a large current density of 100 mA cm⁻² at a small overpotential of 292 mV. Intriguingly, at the high current density of 100 mA cm⁻², the AD-HN-Ir electrocatalyst still exhibits robust stability with a prominent retention of ~80% of the initial current density after 50 h acidic OER operations. Importantly, the SEM image shows that no significant structural collapse was observed in the AD-HN-Ir electrocatalyst.

Figure N5. (a) The OER stability for the AD-HN-Ir electrocatalyst at applied potential of 1.53 V, and (b) SEM image of AD-HN-Ir electrocatalyst after OER reaction.

Accordingly, the Figure N5 has been added in Supplementary Information as the Supplementary Fig. 21a and 21b. In line 14 of page 10 of the revised manuscript, the following text **has been added**: “*Interestingly, the AD-HN-Ir electrocatalyst delivers good activity and stability at a high current density of 100 mA cm⁻² (Supplementary Fig. 21a and 21b).*”

Does the catalyst electrode survive 2000 mA/cm² and 2 V cell potential, which are

current state-of-the-art performance parameters for PEM electrolyzers , e.g. outlined in https://www.energy-x.eu/wp-content/uploads/2020/02/Energy_X_Research-needs-report_final_24.02.2020.pdf

How long do the electrodes last under these more realistic high current/voltage conditions?

Reply: Thank for your nice question. To reach the high current density of 500 mA cm^{-2} for the AD-HN-Ir electrocatalyst, the high potential of 1.75 V was applied on the electrode. The AD-HN-Ir electrocatalyst retains $\sim 50\%$ of the initial current density after 10 h acidic OER operations (Figure N5). The OER performance in acid condition is a great challenge for carbon-based materials because of its easy oxidation at high anodic voltage ($\sim 1.75 \text{ V}$) in an acidic environment (*ChemElectroChem* 5, 583-588 (2018); *Angew. Chem.* 130, 16749-16753 (2018)). Therefore, the well-designed Ir sites were spatially confined in the 3D conductive carbon substrate to display high activity and robust stability during acidic OER process only below the potential of 1.75 V. It is difficult to achieve a high activity and stability at a higher current density of 1000 and 2000 mA cm^{-2} because of oxidation corrosion of carbon substrate under high oxidation potential.

Figure N6. (a) The OER stability for the AD-HN-Ir electrocatalyst at applied potential of 1.75 V, and (b) The SEM image of AD-HN-Ir electrocatalyst after OER reaction.

Accordingly, the Figure N6 has been added into the Supplementary Information as Supplementary Fig. 21c and 21d. In line 20 of page 10 of the revised manuscript, the following text **has been added**: “To further clarify the activity and stability at higher

current densities, a higher current density of 500 mA cm^{-2} was tested for the AD-HN-Ir electrocatalyst. The AD-HN-Ir electrocatalyst exhibits a poor stability with a retention of ~50% of the initial current density (Supplementary Fig. 21c and 21d).” And in reverse line 3 of page 10 of the revised manuscript, the following text **has been added**: “It is difficult to achieve a high activity and stability at higher current densities of 1000 and 2000 mA cm^{-2} because of oxidation corrosion of the carbon substrate under high oxidation potentials.”

Is it possible for the authors to estimate the lifetime under PEM electrolyzer conditions using the S-number (https://www.nature.com/articles/s41929-018-0085-6?WT.feed_name=subjects_hydrogen-energy), e.g. by measurement of Ir loss to the electrolyte?

Reply: Thank you for your constructive suggestion. We agree with you that the stability number (S-number) can be used to evaluate the stability of the active site of Ir-based oxide electrocatalysts (*Nat. Catal.* 1, 508-515 (2018)). According your suggestion, the dissolved content of metal Ir, the calculated S-number and lifetime are shown in Figure N7. Significantly higher S-number and longer lifetime of the AD-HN-Ir electrocatalyst were observed at the current density of 100 mA cm^{-2} . It further proves that the AD-HN-Ir electrocatalyst delivers a good activity and stability at the current density of 100 mA cm^{-2} related to 500 mA cm^{-2} .

As is known, many reported Ir based oxide electrocatalysts have been dissolved and surface structure reconstruction in acidic solution under the voltage driven, thus achieve a high catalytic activity (*Science* 353, 1011–1014 (2016), *Nat. Commun.* 7, 12363 (2016)). The S-number is closely related to the amount of dissolved iridium and the evolved oxygen, which can be used to evaluate the stability of the active site of Ir-based oxide electrocatalysts. In this work, atomically dispersed Ir active sites coupled in 3D carbon substrate with active hetero-Ir-N₄ moieties, which can efficiently avoid the obvious changes of the Ir oxidation state and significant dissolution of Ir active sites. The advanced *in-situ* XAFS and SRIR techniques have revealed that the real active

structure is the O-hetero-Ir-N₄ moiety during OER process. However, a high oxidation potential applied on the electrode causes the oxidation of 3D carbon substrate, and then the dissolution of Ir active sites can be observed. Therefore, the S-number cannot directly reveal the relationship between the structure of Ir active site and the performance of atomically dispersed Ir electrocatalyst system, but it can also be used as a parameter to evaluate the stability of the catalyst. As shown in Figure N7a, the low amount of dissolved iridium at low potential of 1.53 V (100 mA cm⁻²) is due to the stable O-hetero-Ir-N₄ structure. Furthermore, the calculated S-number and lifetime of AD-HN-Ir under lower potential (1.53 V @ 100 mA cm⁻²) are significantly higher than those of higher potential (1.75 V @ 500 mA cm⁻²), suggesting the decay of the stability attributed to the corrosion of the carbon substrate under a high potential.

Figure N7. (a) Dissolved content of metal Ir, (b) the calculated constant S-number under 1.53 and 1.75 V vs. RHE. Error bars were obtained by standard deviation of at least three independent measurements. (c) Calculation of lifetime for AD-HN-Ir electrocatalyst.

Accordingly, the Figure N7 has been added in Supplementary Information as Supplementary Fig. 22. In line 24 of page 10 of the revised manuscript, the following text **has been added**: “Significantly higher S-number and longer lifetime of AD-HN-Ir electrocatalyst were observed under a lower potential of 1.53 V (reach to 100 mA cm⁻²) (the subsequent potentials are all relative to a reversible hydrogen electrode), suggesting that the decay of the stability was attributed to the corrosion of the carbon substrate at a high potential (Supplementary Fig. 22).” In reverse line 4 of page18 of the revised manuscript, the following text **has been added**: “The S-number is defined as the ratio between the amount of evolved oxygen (calculated from Q_{total}) and the amount of dissolved iridium (extracted from ICP-MS data). The lifetime of electrocatalyst was calculated on the basis of the equation:

$$t = \frac{S \times z \times F \times m}{j \times M}$$

where t is the lifetime of the electrocatalyst (s), S is the stability number, z is the number of electrons per evolved O_2 , F is the Faraday constant ($96,485 \text{ C mol}^{-1}$), m is the loaded mass of iridium (g cm^{-2}), j is the applied current density (A cm^{-2}) and M is the molar mass of iridium (192.2 g mol^{-1}).” The following reference of “Geiger, S. *et al.* The stability number as a metric for electrocatalyst stability benchmarking. *Nat. Catal.* **1**, 508-515 (2018).” **have been added** in the revised manuscript as reference 28.

C3 – I find the explanation and data plausible to explain the increase in performance. Yet, the discussion seems to rely on results that are presented later in the text. I comment on it below and recommend to move the discussion after the presentation of all data/figures.

On balance, I find the study potentially suitable for Nature Communications if the claims were better supported as suggested and if there is sufficient novelty relative to a similar publication in this journal (<https://www.nature.com/articles/s41467-019-12886-z>).

Reply: We are greatly grateful to the reviewer for your nice question and careful inspection. In this work, the novelty is mainly reflected in the 3D electrode preparation and mechanism study by *in-situ* technologies. On the one hand, we design a new type of atomically dispersed Ir active sites coupled in hetero-nitrogen-configured 3D carbon substrate as stable electrode for high catalytic activity and long-term durability in acidic medium. On the other hand, the advanced *in-situ* XAFS and SRIR techniques can track the dynamic evolutions of Ir active sites and the reaction intermediates with the change of applied potentials in real time, so as to obtain the real active structure and reaction path, which can be used to reveal the microscopic reaction mechanism of catalytic reaction.

Recently, Cao *et al.* reported that Ru-N-C catalyst delivered an efficient OER performance with 10 mA cm^{-2} at an overpotential of 267 mV and stable operation at 10

mA cm⁻². At the same time, the powder of Ru-N-C sample was coated on the three-dimensional carbon cloth through physical adsorption as the *in-situ* test electrode. Only the active structure of O-Ru-N₄ was observed by *in-situ* characterization technique, but the absence of key reaction intermediates observation cannot be used to intuitively reveal the microscopic reaction mechanism (*Nat. Commun.* 10, 4849 (2019)). However, in this work, we design a new type of atomically dispersed Ir active sites coupled in hetero-nitrogen-configured 3D carbon substrate via a controllable “electric-driven amino-induced” strategy, realizing high catalytic activity and long-term durability even under high current density of 100 mA cm⁻² in acidic medium. Furthermore, the three-dimensional assembled electrode prevents signal interference caused by catalyst sample shedding during the *in-situ* testing. And the advanced *in-situ* XAFS and SRIR techniques indicated that one oxygen atom is dynamically coupled onto the Ir active site and then a key *OOH is produced by the electrophilic effect of dynamic-coupling oxygen over O-hetero-Ir-N₄ moieties under working potentials. The electronic and coordination structure evolution of active metal sites as well as the key reaction intermediate *OOH under different applied-potentials were simultaneously captured experimentally for the AD-HN-Ir electrocatalyst. At the atomic level, we revealed the correlation between the active site structure and the reaction intermediates. And the reaction microscopic mechanism of acid OER four-electron process can be further deeply understood based on the atomically dispersed Ir electrocatalyst.

Please also address the following points:

1) Title

a) The title states “single atom”, yet Fig. 1c is too small to judge and Fig. S5 shows larger spots (near the 5 of the scale bar). How many spots were analyzed? Is the selection of Fig. S5b representative? I find the expression “atomically dispersed Ir” much better than claiming all single atoms, which may not be supported by EXAFS analysis, see below.

Reply: This question is very important to improve the quality of this work. According

to your suggestion, the more appropriate magnification of HAADF-TEM image was provided and shown in the Figure N8. It demonstrates that atomically dispersed Ir atoms were uniformly anchored to the hetero-nitrogen-configured 3D carbon substrate without significant agglomeration. In the Figure N9, more than 30 points were analyzed in HAADF-TEM image. The bright spots corresponded to Ir atoms are uniformly dispersed along with similar dimensions. As shown in the Figure N9b, the bright spots are highly dispersed across the substrate and intensity profiles in different selection were further analyzed, suggesting the atomically dispersed Ir atoms chemically coupled onto hetero-nitrogen-configured 3D carbon substrate with a particle size of $\sim 0.21\text{-}0.23$ nm.

Figure N8. HAADF-TEM image and the inset is TEM image for AD-HN-Ir electrocatalyst.

Figure N9. (a) Atomic-resolution HAADF-STEM image of AD-HN-Ir electrocatalyst, (b) intensity profile along the line 1 in (a) and (c) intensity profile along the line 2 and 3 in (a), indicating uniform distribution of Ir atoms.

Accordingly, the Figure 1c in previous manuscript has been replaced by the Figure N8. The Figure N9 has been added in the revised Supplementary Information as

Supplementary Fig. 5. In line 16, page 5 of the revised manuscript, the following text **has been added**: “*In particular, high-angle annular dark-field scanning transmission electron microscopy (HAADF-STEM) further confirmed that atomically dispersed Ir atoms were riveted on the hetero-nitrogen-configured 3D carbon substrate surface with a particle size of ~0.21-0.23 nm (Supplementary Fig. 5).*”

Moreover, according to your suggestion, to more accurately describe the form of AD-HN-Ir electrocatalyst, the expression “single atom Ir” in the previous manuscript has been replaced by the “atomically dispersed Ir” in the revised manuscript.

b) The study is in situ but not operando. Operando requires a simultaneous product measurement

(e.g. <https://www.sciencedirect.com/science/article/pii/S0920586104008065?via=ihub>). That could be the current for electrocatalysis but no data from in situ experiments is shown. Please either add this data (which must be convincing) or use “in situ”.

Reply: We thank the reviewer for the constructive suggestion that is quite useful for improving the accuracy of this work. According to your suggestion, the “*operando*” in the previous manuscript has been replaced by “*in-situ*” in the revised manuscript.

c) I am not sure what is meant by “dynamic-coupling oxygen”. Oxygen species adsorb and desorb during the catalytic cycle. This is expected. In order to oxidize hydroxide/water, one must adsorb it and then release O₂. I recommend to remove the confusing expression or discuss in detail what new information it adds. Also in the main text, the word “dynamic” seems to have no important meaning and most sentences would contain the same information when it was deleted.

Reply: Thanks for your nice question. We are sorry that the meaning and concept of “dynamic-coupling oxygen” was not clearly expressed in the previous manuscript. In this work, we design a type of atomically dispersed Ir active sites anchored onto the 3D carbon substrate with the hetero-Ir-N₄ moieties. Interestingly, by combining *in-situ* XAFS and S-FTIR technologies, we observed in experiment that one oxygen atom was

dynamically coupled to the Ir active site with the form of the O-hetero-Ir-N₄ structure under working conditions. After removal the working potential, the active structure returns to the hetero-Ir-N₄ structure proved by in-situ SRIR results (Figure N10). Therefore, we use “dynamic-coupling oxygen” to highlight the potential driven oxygen atom coupled to the Ir active site during the reaction process. Furthermore, according to your suggestion, we have checked the whole manuscript carefully and removed some superfluous expressions of “dynamic” without compromising the meaning of the statement in the revised manuscript.

Figure N10. *In-situ* SRIR measurements in the range of 1300–600 cm⁻¹ under 1.45 V and after reaction (A. R.) conditions for AD-HN-Ir electrocatalyst.

Accordingly, the Figure N10 has been added in the revised Supporting Information as Supplementary Fig. 27. In line 14, page 13 of the revised manuscript, the following text **has been added**: “*The dynamically-coupled O disappears after reaction by the SRIR results (Supplementary Fig. 27), suggesting that the coupling of one oxygen atom on the Ir active site is a dynamic process with the change of potentials.*”

2) Please add the source of the Ir atoms to the main text. It was hard to find this important information in the manuscript.

Reply: Thanks for your reminding. In this work, the source of the Ir atoms are mainly

from H_2IrCl_4 via a typical ion exchange strategy. Accordingly, in line 6, page 5 of the revised manuscript, the following text **has been added**: “*It is noted that the surface reductive benzenoid-amine groups and $-\text{NH}_2$ -derived uncoordinated N sites act as anchoring sites for Ir atoms in the H_2IrCl_4 solution via a typical ion exchange strategy.*”

3) EXAFS analysis

a) Please add the details of the Fourier transform and used window function, otherwise the figure is not reproducible.

Reply: It is a good question to improve the quality of this manuscript. The acquired EXAFS data were processed according to the standard procedures using the ATHENA module implemented in the IFEFFIT software packages (*J. Synchrot. Radiat.* 8, 322–324 (2001)). Subsequently, k^2 -weighted $\chi(k)$ data in the k -space ranging from 3.0–11.8 \AA^{-1} were Fourier transformed to real space using a Hanning windows ($dk = 1.0 \text{\AA}^{-1}$) to separate the EXAFS contributions from different coordination shells.

Accordingly, in line 7, page 7 of the Supplementary Information, the following text **has been added**: “***XAFS measurements and fitting:*** *The acquired EXAFS data were processed according to the standard procedures using the ATHENA module implemented in the IFEFFIT software packages (J. Synchrot. Radiat. 8, 322–324 (2001)). Subsequently, k^2 -weighted $\chi(k)$ data in the k -space ranging from 3.0–11.8 \AA^{-1} were Fourier transformed to real space using a Hanning windows ($dk = 1.0 \text{\AA}^{-1}$) to separate the EXAFS contributions from different coordination shells.*”

b) Please include the data in k -space, which contains more information

Reply: It is a nice question. According to your suggestion, the L -edge EXAFS $k^2\chi(k)$ functions of AD-HN-Ir electrocatalyst and reference samples were added in Supplementary Fig. 8 of the revised Supporting Information. As shown in the Figure N10, it can be inferred that the high-valence Ir active site is formed in the first-shell of Ir-N coordination, based on the position of the Ir L_3 absorption edge and the first-shell coordination peak at $\sim 1.6 \text{\AA}$. Figure N11b shows the difference of the EXAFS

oscillation frequency between the AD-HN-Ir electrocatalyst and the reference samples. This phenomenon indicates that the atomically dispersed Ir active sites with a Ir-N coordination configuration of N atoms in the first-shell are anchored to a hetero-nitrogen-configured 3D carbon substrate.

Figure N11. (a) Ir L_3 -edge XANES spectra, (b) $k^2\chi(k)$ curves of Ir L_3 -edge EXAFS oscillation functions and (c) Fourier transforms (FTs) of the Ir L_3 -edge EXAFS oscillations functions for the AD-HN-Ir, Ir-NP/NC, Ir foil, IrO_2 , and the fitting curve of k^2 -weighted EXAFS spectrum of the AD-HN-Ir electrocatalyst.

Accordingly, the Figure 2e and 2f have been replaced by Figure N11a and N11c in the revised manuscript. The Figure N11 has been added in the revised Supplementary Information as Supplementary Fig. 8. In reverse line 8, page 7 of the revised manuscript, the following text **has been added**: “*The L_3 -edge EXAFS function $k^2\chi(k)$ of the AD-HN-Ir electrocatalyst (Supplementary Fig. 8) shows that the atomically dispersed Ir active sites with Ir-N coordination configuration in the first shell were anchored to a hetero-nitrogen-configured 3D carbon substrate.*”

c) P7. What is meant by “oscillation frequency”? The EXAFS oscillations in Fig. 2e? Only one period is shown and it matches that of IrO_2 and not that of a material with Ir-N bonds.

Reply: Thanks for your careful inspection. I am sorry that an ambiguous expression of “oscillation frequency” appeared in the previous manuscript due to the mistake. In fact, we would like to describe the intensity and width of the white line peaks of XANES spectra in Figure 2e, so that the differences surrounding Ir atoms can be distinguished in various coordination environments. The AD-HN-Ir electrocatalyst exhibits a larger

and narrower white peak in comparison with that of Ir foil.

Firstly, in preparation, the surface reductive benzenoid-amine groups and -NH₂-derived uncoordinated N sites can act as the anchoring sites for Ir atoms during ions exchange and pyrolysis processes. Furthermore, according to the XPS result of N 1s, the Ir-N_x species have been formed for the AD-HN-Ir electrocatalyst. Above all, only main peak located at ~1.6 Å can be assigned to the first shell of Ir-N coordination with hetero-Ir-N₄ based on the result of FT curve fitting of AD-HN-Ir electrocatalyst.

Accordingly, the “oscillation frequency” in the previous manuscript has been replaced by “the intensity and width of the white line peak” in the in the revised manuscript, and in line 6, page 7 of the revised manuscript, the following text **has been added**: “Figure 2f shows one main peak located at ~1.6 Å that can be assigned to the first-shell of Ir-N coordination according to the N 1s XPS result as well as the electrophilic centre of N.”

d) How were the fits performed? In k-space in which range or in R-space or Fourier-filtered k-space (Q-space in Artemis)? This is important to reproduce the work and judge its meaningfulness

Reply: We thank the reviewer for the constructive suggestion that is quite useful for improving the quality of this work. The acquired EXAFS data were analyzed according to the standard procedures using the ATHENA module implemented in the IFEFFIT software packages (*J. Synchrot. Radiat.* 8, 322–324 (2001)). Subsequently, k^2 -weighted $\chi(k)$ data in the k -space ranging from 3.0–11.8 Å⁻¹ were fast-Fourier-transformed to the real space using a Hanning windows ($dk = 1.0$ Å⁻¹) to separate the EXAFS contributions from different coordination shells. The amplitude reduction factor S_0^2 (0.78) was obtained from the L_3 -edge EXAFS curve fitting for phthalocyanine iridium. As for the *ex-situ* for AD-HN-Ir electrocatalyst, the curve fitting was done on the k^2 -weighted EXAFS function $\chi(k)$ data in the k -range of 3.0- 11.8 Å⁻¹ and in the R -range of 1.0–2.5 Å. The number of independent points for these samples are $N_{\text{ipt}}=2\Delta k \cdot \Delta R/\pi=2 \times (11.8-3.0) \times (2.5-1.0)/\pi=8$. However, compared with that of the *ex-situ* sample, the first

coordination peaks of *in-situ* AD-HN-Ir electrocatalyst under 1.25, 1.35 and 1.45 V show an obviously increased strength and a higher R shift, which was ascribed to the addition of Ir-O coordination. Therefore, the two subshells of Ir-N and Ir-O coordination were considered for the curve fitting of AD-HN-Ir electrocatalysts under work conditions. During curve-fittings, each of the Debye–Waller factors (σ^2), coordination numbers (N), interatomic distances (R) and energy shift (ΔE_0) was treated as adjustable parameters. Notably, in order to reduce the number of adjustable fitting parameters, the N and ΔE_0 of Ir-N coordination shell were fixed and equal to the parameters of the *ex-situ* sample.

Accordingly, in line 16, page 7 of the Supplementary Information, the following text **has been added**: “**XAFS measurements and fitting**: “The amplitude reduction factor S_0^2 (0.78) was obtained from the L_3 -edge EXAFS curve fitting for phthalocyanine iridium. As for the *ex-situ* for AD-HN-Ir electrocatalyst, the curve fitting was done on the k^2 -weighted EXAFS function $\chi(k)$ data in the k -range of 3.0- 11.8 \AA^{-1} and in the R -range of 1.0–2.5 \AA . The number of independent points for these samples are $N_{\text{ipf}}=2\Delta k \cdot \Delta R/\pi=2 \times (11.8-3.0) \times (2.5-1.0)/\pi=8$. However, compared with that of the *ex-situ* sample, the first coordination peaks of *in-situ* AD-HN-Ir electrocatalyst under 1.25, 1.35 and 1.45 V show an obviously increased strength and a higher R shift, which was ascribed to the addition of Ir-O coordination. Therefore, the two subshells of Ir-N and Ir-O coordination were considered for the curve fitting of AD-HN-Ir electrocatalysts under work conditions. During curve-fittings, each of the Debye–Waller factors (σ^2), coordination numbers (N), interatomic distances (R) and energy shift (ΔE_0) was treated as adjustable parameters. Notably, in order to reduce the number of adjustable fitting parameters, the N and ΔE_0 of Ir-N coordination shell were fixed and equal to the parameters of the *ex-situ* sample.”

e) What structural model was used and what were the parameters to obtain the phase functions?

Reply: It is a good question, and we are sorry that the structural models and the

parameters were not stated clearly in the previous manuscript. For the L_3 -edge EXAFS curves fitting of first-shell of Ir-N coordination, the phthalocyanine iridium was selected as the fitting structural model. And the iridium oxide can be selected to fit the first-shell of Ir-O coordination for the AD-HN-Ir electrocatalyst.

To obtain the detailed structural parameters around the Ir active sites in the as-prepared samples, quantitative curve fittings were carried out for the Fourier-transformed $k^2 \chi(k)$ in R-space using the ARTEMIS module of IFEFFIT. The amplitude $F_i(k, R_i)$ and phase shifts $\Phi_i(k, R_i)$ functions for the first-shell of Ir-N/O coordination were calculated with the FEFF8 code in the muffin-tin (MT) self-consistent-field approximation. Default values of MT radii and muffin-tin radii overlap of 15% between contiguous spheres were used to simulate the atomic bond. To account for energy dependent exchange-correlation potential, the energy- and position-dependent optical Hedin-Lundqvist potential was employed. (*Rev. Mod. Phys.* **72**, 621 (2000), *Phys. Rev. B* **76**, 174107 (2007)). The calculations were performed for a cluster with radius of 8 Å having the structure of phthalocyanine iridium (iridium oxide) and centered at the Ir-N (Ir-O) atom, respectively.

Accordingly, in reverse line 1, page 7 of the Supplementary Information, the following text **has been added**: “*Quantitative curve fittings were carried out for the Fourier transformed $k^2\chi(k)$ in R-space using the ARTEMIS module of IFEFFIT. The amplitude $F_i(k, R_i)$ and phase shifts $\Phi_i(k, R_i)$ functions for the the first-shell of Ir-N/O coordination were calculated with the FEFF8 in the muffin-tin (MT) self-consistent-field approximation. Default values of MT radii and muffin-tin radii overlap of 15% between contiguous spheres were used to simulate the atomic bond. To account for energy dependent exchange-correlation potential, the energy- and position-dependent optical Hedin-Lundqvist potential was employed. The calculations were performed for a cluster with radius of 8 Å having the structure of phthalocyanine iridium (iridium oxide) and centered at the Ir-N (Ir-O) atom, respectively.*”

f) The red line in Fig. 2f has peaks at higher R. Can they be fit with Ir-Ir interactions

(thus indicating no single atom Ir)? why should the peaks in the FT around 3 Å not be significant? They look well above the noise level and fall into the range expected for Ir-Ir distances.

Reply: We thank the reviewer for the insightful suggestion that is quite useful for improving the consistency of this work. In order to identify the source of the high R peak, we tried several k ranges for FT and multi-shells fitting, and it reveals that the peak at 2.74 Å may come from the contribution of a small amount of the first Ir-Ir coordination shell (less than 10%).

Firstly, in order to realize the best background removal for EXAFS data analysis, we selected the different R_{bkg} values of 0.9, 1.0 and 1.1 Å to remove low frequency noise. As shown in Figure N12b, the best background removal was at the $R_{\text{bkg}}=1.1$ Å, and the low frequency noise was removed fully but the high frequency noises are almost the same for the different R_{bkg} values. Furthermore, it can be found that the larger noises appear in the k region of 8 and 12.5 Å⁻¹ (Figure N12a). Subsequently, the four different k ranges were selected for the fast Fourier transform, and the results were shown in Figure N12c. It can be observed that the 2.74 Å FT peak intensity of the 3.0-10.2 Å⁻¹ region is reduced by 100% in comparison with that of 3.0-11.8 Å⁻¹ region, suggesting the increase of the high-frequency noise in the high k region of 3.0-11.8 Å⁻¹ because of the low Ir content for fluorescence XAFS measurement. However, the large k region of 3.0-11.8 Å⁻¹ was necessary for more independent free points ($N_{\text{pt}}=2\Delta k \cdot \Delta R/\pi=2 \times (11.8-3.0) \times (2.5-1.0)/\pi=8$) for the better data curve fitting. Finally, the FT spectrum in Figure N12d shows that the peaks at 2.59 and 2.85 Å can be attributed to the first Ir-Ir shell (Ir foil) and the second Ir-O-Ir shell (IrO₂), respectively. Noted that the FT peak at 2.74 Å for the AD-HN-Ir electrocatalyst is obviously different from that of the Ir-Ir (Ir-O-Ir) coordination shell of Ir foil (IrO₂), but the existence of Ir-Ir bond cannot be directly excluded. Therefore, the EXAFS spectrum for the AD-HN-Ir electrocatalyst under 1.45 V was fitted by considering of two subshells of first-shell of Ir-N/O bond and the first shell of Ir-Ir bonds. As shown in Figure N12e and N12f, the coordination number of the first shell of Ir-Ir coordination is 0.9, suggesting that Ir-Ir coordination is less than 10%.

Above results significantly prove that 90% Ir atoms were atomically dispersed on 3D carbon-nitrogen substrate with hetero-Ir-N₄ moieties. Therefore, the FT curves after background removal proceeded at the $R_{\text{bkg}}=1.1 \text{ \AA}$ for the AD-HN-Ir electrocatalyst were refitted only considering first-shell of Ir-N/O coordination, because a small amount (10%) of Ir-Ir bonds does not affect the conclusion of this manuscript.

Figure N12. (a) $k^2\chi(k)$ curves of Ir L -edge EXAFS oscillation functions. Corresponding k^2 -weighted FT of Ir L_3 -edge EXAFS oscillation functions (b) under different R_{bkg} values of 0.9, 1.0 and 1.1 \AA for background removal and (c) at four different k ranges for AD-HN-Ir electrocatalyst. (d) FT spectrum of Ir L_3 -edge EXAFS oscillation functions for AD-HN-Ir, Ir foil and IrO₂. (e) The fitting curve of k^2 -weighted EXAFS spectrum and (f) the $\text{Re}(k^2\chi(k))$ oscillation curve for AD-HN-Ir electrocatalyst under 1.45 V.

Table N2. Structural parameters for the AD-HN-Ir electrocatalyst extracted from quantitative EXAFS curve-fitting by considering the first-shell of Ir-N/O and Ir-Ir coordination.

Sample	Path	N	R (Å)	$\sigma^2(10^{-3}\text{Å}^2)$	$\Delta E_0(\text{eV})$	R-factor
AD-HN-Ir	Ir-N ₁	2.1 ± 0.2	1.95 ± 0.02	2.3 ± 0.4	7.8	0.01
	Ir-N ₁	2.0 ± 0.2	2.03 ± 0.02	2.2 ± 1.1		
	Ir-O	1.9 ± 0.2	2.06 ± 0.02	2.7 ± 1.1		
	Ir-Ir	0.9 ± 0.3	2.78 ± 0.02	6.1 ± 1.1		

Accordingly, the Figure N12 and Table N2 have been added in the revised Supplementary Information as Supplementary Fig. 28 and Supplementary Table 5. In line 11, page 7 of the revised manuscript, the following text **has been added**: “*The peaks at higher R can be attributed to higher frequency noise and a small amount (10%) of Ir-Ir bonds because of the low Ir content for fluorescence XAFS measurement.*”

g) Is a better background subtraction possible for the data in Fig. 5b? Can it be excluded that the difference in the analyzed peak height is due to the distortion that causes the peak at 1 Å? The k-space data would help experts to identify issues with the data and judge if the analysis of the peak at 1.6 Å is sound.

Reply: We are greatly grateful to the reviewer for your nice question and careful inspection. In order to obtain a reliable EXAFS results, one of important matters is to perform background removal and data standardization for the EXAFS data analysis. The background subtraction is to fit the smooth part with a polynomial by the least square method. The background function obtained by the fitting is easy to deduct the background part of slow change from $\mu(E)$. As we all known, the values of R_{bkg} and k -

weight in the background subtraction will slightly affect the FT results. The best background removal was at the $R_{\text{bkg}}=1.1 \text{ \AA}$ (Figure N12b), and the low frequency noise was removed fully but the high frequency noises are almost the same for the different R_{bkg} values. Furthermore, the difference of L-edge EXAFS $k^2\chi(k)$ functions and the FT curves for the AD-HN-Ir electrocatalyst under different applied-potentials further prove dynamic evolution of Ir active sites during OER process (Figure N13).

Figure N13. (a) $k^2\chi(k)$ curves of Ir L-edge EXAFS oscillation functions Corresponding k^2 -weighted FT of Ir L_3 -edge EXAFS oscillation functions for the AD-HN-Ir electrocatalyst under different applied-potentials.

Accordingly, the Figure 5b in the previous manuscript has been replaced by the FigureN13b in the revised manuscript. In line 14, page 7 of the revised Supplementary Information, the following text **has been added**: “The best background removal was at the $R_{\text{bkg}}=1.1 \text{ \AA}$, and the low frequency noise was removed fully.” In line 3, page 15 of the revised manuscript: “The difference of L-edge EXAFS $k^2\chi(k)$ functions for the AD-HN-Ir electrocatalyst under different applied-potentials further prove dynamic evolution of Ir active sites during OER process.”

4) Activity

a) Please test the activity and stability also at higher current densities such as 500 mA/cm², 1000 mA/cm² and 2000 mA/cm² if possible. PEM electrolyzers are operated under harsher conditions and I am not convinced stability at 10 mA/cm² is relevant for this application.

Reply: We thank the reviewer for the constructive suggestion that is quite useful for

improving the quality of this work. According to your suggestion, we performed the activity and stability of AD-HN-Ir electrocatalyst at higher densities such as 100 and 500 mA cm⁻². As shown in the Figure N14, the AD-HN-Ir electrocatalyst reaches a large current density of 100 mA cm⁻² at a small overpotential of 292 mV. Intriguingly, at the high current density of 100 mA cm⁻², the AD-HN-Ir electrocatalyst still exhibits robust stability with a prominent retention of ~80% of the initial current density after continuous 50 h acidic OER operations. However, to reach the high current density of 500 mA cm⁻² for the AD-HN-Ir electrocatalyst, a high potential of ~1.75 V was applied on the electrode. The AD-HN-Ir electrocatalyst exhibits poor stability with a retention of ~50% of the initial current density after continuous 10 h acidic OER operations. OER in acid solution is also a big challenge for carbon-based materials because of easily oxidized carbon-based materials at high anodic voltage (1.75 V) in an acidic environment (*ChemElectroChem* 5, 583-588 (2018); *Angew. Chem.* 130, 16749-16753 (2018)). Therefore, the well-designed Ir active sites were spatially confined in the 3D conductive carbon substrate to display high activity and robust stability during acidic OER process below the potential of <1.75 V. It is difficult to achieve high activity and stability at higher current densities of 1000 and 2000 mA cm⁻² because of oxidation corrosion of carbon substrate under high oxidation potentials.

Figure N14. OER stability for the AD-HN-Ir electrocatalyst at applied potentials of 1.53 V (a) and 1.75 V (c). SEM image (b), (d) of the AD-HN-Ir electrocatalyst after reaction.

Accordingly, the Figure N14 has been added in the revised Supplementary Information as Supplementary Fig. 21. In line 14, page 10 of the revised manuscript, the following text **has been added**: “*Interestingly, the AD-HN-Ir electrocatalyst delivers good activity and stability at a high current density of 100 mA cm⁻² (Supplementary Fig. 21a and b).*” In line 20, page 10 of the revised manuscript, the following text **has been added**: “*To further clarify the activity and stability of higher current densities, the stability of a higher current density of 500 mA cm⁻² at a higher potential of 1.75 V was tested for the AD-HN-Ir electrocatalyst. The AD-HN-Ir electrocatalyst exhibited poor stability with a retention of ~50% of the initial current density (Supplementary Fig. 21c and d). It is difficult to achieve high activity and stability at the higher current densities of 1000 and 2000 mA cm⁻² because of oxidation corrosion of the carbon substrate under high oxidation potentials.*”

b) Please repeat at least the activity measurements to get statistics.

Reply: Thank you for your constructive suggestion. The results of activity measurements are shown after repeated measurements. According your suggestion, the repeat activity measurements were shown in the Figure N15. It reveals that the overpotentials under 10 mA cm^{-2} and 100 mA cm^{-2} of AD-HN-Ir electrocatalyst are significantly lower than those of the reference and standard samples after repeated performance tests. Furthermore, the statistical results of mass activity further prove the reproducibility of high activity of the AD-HN-Ir electrocatalyst.

Figure N15. (a) Overpotentials at 10 and 100 mA cm⁻² for AD-HN-Ir, Ir-NC, IrO₂ and Ir-NP/NC. (b) Mass activity (M.A.) and turnover frequency (TOF) at overpotentials of 216 and 260 mV for AD-HN-Ir and IrO₂.

Accordingly, the Figure 3b and 3e in the previous manuscript have been replaced by the Figure N15a and N15b in the revised manuscript. In line 18, page 9 of the revised manuscript, the following text **has been added**: “Notably, the mass activity and TOF of AD-HN-Ir are approximately $2860 \text{ A g}_{\text{metal}}^{-1}$ and 5110 h^{-1} , respectively, at a small overpotential of 216 mV, ~480 and 510 times larger than those of the commercial IrO₂ ($6 \text{ A g}_{\text{metal}}^{-1}$, 10 h^{-1}) by performance test statistics.”

c) It is good to perform measurements in a two-electrode cell but the one used by the authors is by no means close enough to a PEM electrolyzer system make statements about “practical electrolyzers”

Reply: We thank the reviewer for the constructive suggestion that is quite useful for improving the quality of this work. According to your suggestion, a PEM electrolyser system was selected to assess the practical application capability of the AD-HN-Ir

electrocatalyst (Figure N16). The resultant electrolyser delivered 500 mA cm^{-2} at $\sim 1.75\text{-}1.85 \text{ V}$ over continuous 10 h operations, indicating a degree of stability of the AD-HN-Ir electrocatalyst. However, the stability of the AD-HN-Ir electrocatalyst decreased significantly under a high current of 1000 mA cm^{-2} attributed to the significant oxidation of the carbon substrate.

To capture the industrial potential of the AD-HN-Ir electrocatalyst, we also carried out the stability of the AD-HN-Ir electrocatalyst at high current densities by a PEM electrolyser system. As shown in Figure N16, the 3D carbon electrode is bonded to the surface of the proton exchange membrane, while commercial Pt/C is sprayed on the other side of the proton exchange membrane as a cathode to form the membrane electrode (Figure N16a). The PEM electrolyser system was tested under simulated industrial conditions ($80 \text{ }^\circ\text{C}$). It can be drawn from Figure N16b that the input potential of water splitting in PEM electrolyser system is about $1.75\text{-}1.85 \text{ V}$ for achieving the current density of 500 mA cm^{-2} , suggesting a degree of stability attributed to stable O-hetro-Ir-N₄ structure. However, to maintain the higher current density of 1000 mA cm^{-2} , the required voltage is significantly increased from 1.95 to 2.7 V as the reaction time goes on, revealing that the stability of the catalyst rapidly attenuates at high current density due to the significant oxidation of the carbon substrate at high potentials.

Figure N16. (a) Membrane electrode assembly. (b) The cell voltages of the PEM electrolyzer system held at 500 and 1000 mA cm⁻². (c) The PEM electrolyzer cell and (d) a PEM electrolyzer system.

Accordingly, the Figure N16 has been added in the revised Supplementary Information as Supplementary Fig. 23. In reverse line 1, page 10 of the revised manuscript, the following text **has been added**: “*To capture the industrial potential of the AD-HN-Ir electrocatalyst, we also carried out the stability of the AD-HN-Ir electrocatalyst at high current density by a PEM electrolyser system under simulated industrial conditions (80 °C, Supplementary Fig. 23). The resultant electrolyser delivered 500 mA cm⁻² at ~1.75-1.85 V over 10 h operation, indicating a degree of stability of the AD-HN-Ir electrocatalyst. However, the stability of the electrocatalyst decreased significantly at a high current of 1000 mA cm⁻², which is attributed to the significant oxidation of the carbon substrate.*”

d) P15. 100VmA is not a universally suitable potential window for Cdl determination by cyclic voltammetry and the range of sweep speeds is rather narrow

(<https://iopscience.iop.org/article/10.1088/2515-7655/abee33>). This can introduce large systematic errors in the determination of specific activity (in Fig. S11).

Reply: Thank you for your nice questions. We really compliment the reviewer for your expert knowledge in the field of electrochemistry. There is a close relationship between double layer capacitance (C_{dl}) and potential window and sweep speeds of cyclic voltammetry (*J. Phys. Energy* 3, 034013 (2021)). According to your suggestion, the more suitable potential window (300 mV) and the wider range of sweep speeds (0.01-1.0 V/s) were selected to perform cyclic voltammetry (CV) tests. As shown in the Figure N16, the C_{dl} can be calculated as 5.2 mF from the cyclic voltammetry tests.

As is known, in order to obtain a reliable C_{dl} by the CV tests, it is necessary to select measuring windows larger than 100 mV not only to ensure reaching a steady charging current, but also to ensure that i_c values extracted from the anodic and cathodic scans are similar (*J. Phys. Energy* 3, 034013 (2021)). In this work, we performed CV tests at a wide redox process-free window of 300 mV to endure steady charging current and similar anodic and cathodic scans. Furthermore, to achieve a small deviation and sufficient potential points, it is recommended to use a scan rate range as wide as possible to reach suitable current values. On the one hand, it needs to avoid too slow scanning speed to make the experiment unnecessarily time-consuming; on the other hand, it also needs to avoid too large scan rates that may lead to large deviations originating from potentiostat limitations. Therefore, in this work, the suitable sweep speeds of 0.01, 0.05, 0.1, 0.5 and 1.0 V/s were selected to perform CV tests. It can be seen from Figure N15, the C_{dl} of AD-HN-Ir electrocatalyst can be calculated as 5.2 mF by the CV tests, slightly larger than those of Ir-NC and IrO₂ (4.1 and 3.7 mF, Figure N17 and N18). And the electrochemically active surface area (ECSA) of AD-HN-Ir electrocatalyst was obtained by roughness factor, and the specific activity of AD-HN-Ir electrocatalyst still surpasses those of Ir-NC and IrO₂ when normalizing the current density to per ECSA (Figure N19).

Figure N17. Double-layer capacitance measurements. (a), (c) CVs were conducted in a non-Faradaic region of voltammogram at the following scan rate: 0.01, 0.05, 0.1, 0.5, 1 V s^{-1} . (b), (d) The difference in charging currents variation at an underpotential plotted against scan rate for estimation of double-layer capacitance (C_{dl}).

Figure N18. Double-layer capacitance measurements. (a) CVs were conducted in a non-Faradaic region of voltammogram at the following scan rate: 0.01, 0.05, 0.1, 0.5, 1 V s^{-1} . (b) The difference in charging currents variation at an underpotential plotted against scan rate for estimation of double-layer capacitance (C_{dl}).

Figure N19. (a) OER polarization curves of AD-HN-Ir, Ir-NC and IrO₂ in O₂-saturated 0.5 M H₂SO₄ based on the ECSA.

Accordingly, the Supplementary Fig. 12, 13 and 14 in the previous manuscript have been replaced by the Figure N17, N18 and N19 in the revised Supplementary Information. In reverse line 1, page 17 of the revised manuscript, the following text **has been added**: “The double layer capacitance (C_{dl}) and the electrochemically active surface area (ECSA) can be calculated by CV tests. It is necessary to select measuring windows larger than 100 mV not only to ensure reaching a steady charging current but also to ensure that the i_c values extracted from the anodic and cathodic scans are similar. Furthermore, it is recommended to use a scan rate range as wide as possible to reach suitable current values. Therefore, we performed CV tests at a wide redox process-free window of 300 mV and suitable sweep speeds of 0.1, 0.25, 0.50, 0.75 and 1.00 V/s.” The following reference of “Morales, D. *et al.* Seven steps to reliable cyclic voltammetry measurements for the determination of double layer capacitance. *J. Phys. Energy* 3, 034013 (2021).” **have been added** in the revised manuscript as reference 34.

5) At what current density are the Tafel slopes evaluated? Please state explicitly. There seem to be at least two Tafel slopes in the plot of each trace and the traces are not very

linear in this plot

Reply: It is a nice question. As is known, the Tafel slope is an important parameter to evaluate the reaction kinetics of electrocatalysts (*J. Electrochem. Soc.* 160, 142 (2013)), which can be used to assess the electron transport capacity. In the previous manuscript, in order to obtain electron conductivity and electron transfer kinetics of the electrocatalysts, Tafel slopes were evaluated under density of 1-10 mA cm⁻². However, the calculated Tafel slope is not linear for the highly active catalyst at low current where the solid-liquid interface is controlled by the mixture of telephone transmission and substance diffusion. More accurately, the overpotentials interval with the current density of 10-100 mA cm⁻² was re-selected to evaluate the Tafel slopes as shown in Figure N19.

Exactly, the trace of Tafel slopes usually remains linear to describe the kinetics of the electron transfer. It is noted that the reverse reaction cannot be ignored under low overpotential and will cause the trace to deviate from the linearity. The calculated Tafel slope is not linear for the highly active electrocatalyst at low current density (1~10 mA cm⁻²). Therefore, overpotentials interval with the current density of ~10~100 mA cm⁻² was selected to evaluate the Tafel slopes. Tafel slopes in Figure N20 show a good linear relationship and the AD-HN-Ir electrocatalyst reveals the fastest electron transfer kinetics.

Figure N20. Tafel slopes for AD-HN-Ir, Ir-NC, IrO₂ and Ir-NP/NC electrocatalysts.

Accordingly, the Figure 3c in the previous manuscript has been replaced by the Figure N20 in the revised manuscript. In line 10, page 9 of the revised manuscript, the following text **has been added**: “Moreover, Figure 3c displays a smaller Tafel slope of 39 mV dec⁻¹, suggesting a faster OER kinetics and electron transfer occurred over Ir active sites.”

6) It was not quite clear to me during the first read why the catalyst electrode is compared to PANI-Ir. Please revise the text with a statement that the catalyst Ir-NC is based on PANI-Ir but has an additional step in the preparation procedure.

Reply: This is a good question. As the reviewer said, the general process of synthesis for PANI-Ir is similar to that of the preparation of AD-HN-Ir electrocatalyst. However, it is noted that PANI was not treated with concentrated ammonia water for substrate surface functionalization, and the final electrocatalyst did not form hetero-nitrogen coordination. Therefore, it can be called Ir-NC electrocatalyst as a reference sample. According to your suggestion, the “PANI-Ir-800C” in the previous manuscript has been replaced by “Ir-NC” in the revised manuscript.

7) P10. How can R_{ct} be used to represent the H_2O/OOH ion adsorption resistance? R_{ct} relates to charge transfer across an interface. Does the statement make use of the SRIR results discussed later? The connection between R_{ct} and “adsorption kinetics” is likewise unclear. Please revise. In the first half of P10, there is no data yet that OOH is created at all.

Reply: Thanks for your nice question. The R_{ct} is related to the kinetics of the interfacial charge transfer reaction. In accordance with the work of Harrington and Conway (*Electrochem. Acta.*, 32, 1703 (1987)), R_{ct} cannot be interpreted simply as the charge transfer resistances of the electro-adsorption and electro-desorption steps, but are attributed to the properties of more steps including kinetics of adsorption of reactive species in the overall reaction (*J. Electrochem. Soc.*, 160, H142(2013)). Recently, *in-situ* EIS measurements were performed to track the evolution of the adsorbed OH^* intermediates during OER by the R_{ct} fitting (*J. Am. Chem. Soc.* 142, 12087 (2020)). In this work, we also performed *in-situ* EIS measurements to access to the kinetics of adsorption for oxygen-containing reactive species. The kinetics of reactive species adsorption on the electrode surface would significantly affect the kinetics of interfacial electron transfer, leading to significant changes in the fitted R_{ct} . So the R_{ct} can be used to represent the oxygen-containing reactive species ion adsorption kinetics.

According to your suggestion, to improve the logic of the description, the “ H_2O/OOH ” in the previous manuscript have been replaced by “oxygen-containing reactive species” in the revised manuscript. The later *in-situ* SRIR results reveal that key *OOH intermediates were generated over Ir active sites, which further verify the results of *in-situ* EIS measurements.

8) P14. Conclusion: “Hence, this discovery of active sites evolution under working conditions can provide a coordination-engineered strategy for designing advanced acidic-OER electrocatalysts.” What is meant? How exactly are the reported findings useful for other electrocatalysts? What should other scientists look for in good OER catalysts in acid? Finally, what is meant by “active sites evolution”? Species must

adsorb at the active site? Does this refer to the O adsorption on the other side? In the latter case, does it desorb again at lower potential or stay there? The insight that the active state of electrocatalysts differs from the as-made material is not novel and was reported numerous times.

Reply: We are greatly grateful to the reviewer for your nice question and careful inspection. I agree with you that the description in the previous manuscript was not rigorous enough. For the sake of accuracy, the sentence of “*Hence, this discovery of active sites evolution under working conditions can provide a coordination-engineered strategy for designing advanced acidic-OER electrocatalysts.*” in the previous manuscript **has been removed** in the revised manuscript.

As for the evolution of the active sites, in this work, the *in-situ* XAFS and SRIR reveal one oxygen atom coupled on the Ir active site with O-hetero-Ir-N₄ under low potential of 1.25 V. Most importantly, the intensity of vibration absorption showed a positive correlation with the further increase of applied-potentials and remain unchanged when the potential exceeds 1.35 V. This revealed that the active O-hetero-Ir-N₄ structure formed by oxygen coupling was the real active site, and this stable active structure can be maintained during the oxidation reaction to accelerate the OER reaction kinetics. Interestingly, when the applied voltage is removed for a period of time, the SRIR results show that the original absorption vibration peak of Ir-O disappears (Figure N21). This result suggests that the original dynamically-coupled O during the reaction has been desorbed when the OER is stopped. Above results reveal that the coupling of one oxygen atom on the Ir active site is a dynamic process with the change of voltage.

Figure N21. *In-situ* SRIR measurements in the range of 1300–600 cm^{-1} under 1.45 V and after reaction (A. R.) conditions for AD-HN-Ir electrocatalyst.

Accordingly, the Figure N21 has been added in the revised Supplementary Information as Supplementary Fig. 27. In line 14, page 13 of the revised manuscript, the following text **has been added**: “*The dynamically-coupled O disappears after reaction by the SRIR results (Supplementary Fig. 27), suggesting that the coupling of one oxygen atom on the Ir active site is a dynamic process with the change of potentials.*”

9) The panel labels a,b,c,... seem to be missing for most of the figures.

Reply: We are greatly grateful to the reviewer for your nice question and careful inspection. We have examined the entire manuscript carefully and the missing panel labels have been added in the revised manuscript.

10) P17. How was the data smoothed? What method? What was the difference between neighboring steps? 2 cm^{-1} ? 100 points could smooth the spectra too much. Please also show the unprocessed data.

Reply: We thank the reviewer for the constructive suggestion that is quite useful for improving the consistency of this work. The *in-situ* SRIR data were processed by OPUS

software. The curve of an electrolyte without voltage is used as background, and new peaks that appear in the curves after the background subtraction represent the absorption of new species vibration. To obtain a spectral curve with a high SNR, the curves of *in-situ* SRIR were smoothed through 10 points, meaning that 10 consecutive points are averaged to improve the SNR of the curves. Moreover, each high-resolution infrared absorption spectrum with resolution of 2 cm^{-1} was obtained by averaging 514 scans. It suggests that the difference between neighboring steps was $\sim 2\text{ cm}^{-1}$. To further clarify the differences before and after data smoothing, the original spectra without smoothing are presented in Figure N22. The SNR of the smoothed curve is improved to a certain extent, and the new peaks and change trend of the curves are consistent with the original curves, which means that the smoothed curves are reliable to prove the appearance of key new species under working conditions.

Figure N22. *In-situ* SRIR measurements (a) with smoothing and (b) without smoothing in the range of $1300\text{--}600\text{ cm}^{-1}$ under various potentials for AD-HN-Ir electrocatalyst during the OER process.

Accordingly, the Figure N22 has been added in the revised Supplementary Information as Supplementary Fig. 25. In line 1, page 20 of the revised manuscript, the following text **has been added**: “Notably, to show a higher signal-to-noise ratio of the *in-situ* SRIR data, the curves were smoothed through 10 points in the data processing, meaning that 10 consecutive points were averaged to improve the SNR of the curves.”

11) Please have the manuscript checked by a native English speaker. It is largely understandable but some sentences do not read well (e.g. The N K-edge in Figure 2b,

the peaks of excitation of C-N and hetero-Ir-N were obtained.)

Reply: Thanks for your constructive suggestion. The language of the revised manuscript has been polished by ACS Authoring Services.

Editing Certificate

This document certifies that the manuscript

Operando spectroscopic observation of dynamic-coupling oxygen on atomically-dispersed iridium catalyst for acidic water oxidation

prepared by the authors

Hui Su, Wanlin Zhou, Wu Zhou, Yuanli Li, Lirong Zheng, Hui Zhang, Meihuan Liu, Xiuxiu Zhang, Xuan Sun, Yanzhi Xu, Fengchun Hu, Jing Zhang, Tiandou Hu, Qinghua Liu, and Shiqiang Wei

was edited for proper English language, grammar, punctuation, spelling, and overall style by one or more of the highly qualified native English speaking editors at ACS.

This certificate was issued on **August 17, 2021** and may be verified on the ACS website using the verification code **CD14-C30C-2C09-8D5F-31D0**.

Neither the research content nor the authors' intentions were altered in any way during the editing process. Documents receiving this certification should be English-ready for publication; however, the author has the ability to accept or reject our suggestions and changes. To verify the final ACS edited version, please visit our verification page at secure.authoringservices.acs.org/certificate/verify.
If you have any questions or concerns about this edited document, please visit ACS support.

REVIEWERS' COMMENTS

Reviewer #1 (Remarks to the Author):

The authors addressed all my concerns satisfactorily. I recommend it published in Nat. Comm. as its current version.

Reviewer #3 (Remarks to the Author):

Su et al. have extensively revised their manuscript, which improved the overall quality and further strengthened the claims. My concerns were very well addressed. The experimental description has been improved and statistics were added.

I see the main achievements of the authors in reducing its loading while maintain high activity and stability, which is on the roadmap of many international initiatives. In the revised manuscript, the stability is also evaluated at higher current densities. The composite catalyst is less stable at higher current densities (as expected) but the authors give plausible directions for further improvement. Finally, the explanation why their catalyst is more active than simply putting Ir on PANI has been further strengthened by the text revisions. For these reasons, I support publication of the revised manuscript in Nature Communications.

I have two minor remarks

- 1) Suppl. Fig. 11 and text on top of P9: the overpotential depends on temperature because the free energy of water decreases with temperature. 1.23 V is 0 V overpotential only at 25 °C.
- 2) Ref. 28 and 35 are identical.

Report by Marcel Risch

Response to Reviewers' Comments

We are grateful to the reviewers' comments and the editor's decision that our manuscript NCOMMS-21-20746A can be published in a suitably revised version in *Nature Communications*. We have read the reviewers' comments seriously. The detailed replies to the comments are presented in a point-to-point manner as follows:

Reviewer #1 (Remarks to the Author):

The authors addressed all my concerns satisfactorily. I recommend it published in Nat. Comm. as its current version.

Author Reply: Thank you for your constructive criticisms and suggestions, following which the quality of this work has been significantly improved.

Reviewer #3 (Remarks to the Author):

Su et al. have extensively revised their manuscript, which improved the overall quality and further strengthened the claims. My concerns were very well addressed. The experimental description has been improved and statistics were added.

I see the main achievements of the authors in reducing its loading while maintain high activity and stability, which is on the roadmap of many international initiatives. In the revised manuscript, the stability is also evaluated at higher current densities. The composite catalyst is less stable at higher current densities (as expected) but the authors give plausible directions for further improvement. Finally, the explanation why their catalyst is more active than simply putting Ir on PANI has been further strengthened by the text revisions. For these reasons, I support publication of the revised manuscript in Nature Communications.

Author Reply: Thank you for your constructive criticisms and suggestions, following which the quality of this work has been significantly improved.

I have two minor remarks

1) Suppl. Fig. 11 and text on top of P9: the overpotential depends on temperature

because the free energy of water decreases with temperature. 1.23 V is 0 V overpotential only at 25 °C.

2) Ref. 28 and 35 are identical.

Report by Marcel Risch

Question 1. Suppl. Fig. 11 and text on top of P9: the overpotential depends on temperature because the free energy of water decreases with temperature. 1.23 V is 0 V overpotential only at 25 ° C.

Author Reply: Thank you for your careful consideration. According for your suggestion, considering the effect of temperature on the standard electrode potential for water oxidation, the potential has a linear relationship with temperature: $E=E_0+kT$ ($k=0.0001$ V/K). Therefore, the potential of 1.2245 V is the standard water oxidation potential at 80 °C, so the overpotentials under different temperatures have been corrected in the Figure N1. The OER activity is quite temperature-dependent, and the overpotentials of AD-HN-Ir electrocatalyst decrease from 216 to 204 mV at a current density of 10 mA cm^{-2} with the increase of temperature to 80 °C.

Supplementary Fig. 11. Linear sweep voltammetry (LSV) curves of AD-HN-Ir electrocatalyst under 25, 50 and 80 °C in 0.5 M H₂SO₄.

Accordingly, the previous Supplementary Fig.11 has been replaced by the Figure N1 in the revised Supplementary Information. In the line 4, page 7 of the revised

manuscript, the following text **has been added**: “*The OER activity is quite temperature-dependent, and the overpotentials of AD-HN-Ir electrocatalyst decrease from 216 to 204 mV at a current density of 10 mA cm⁻² with the increase of temperature to 80 °C, , which is consistent with what has been observed that a higher temperature provides faster OER kinetics and better OER activity (Supplementary Fig. 11).*”

Question 2. Ref. 28 and 35 are identical.

Author Reply: Thanks for your reminder. Ref. 35 has been deleted in the revised manuscript.